# Effects of canagliflozin on growth and metabolic reprograming in hepatocellular carcinoma cells: Multi-omics analysis of metabolomics and absolute quantification proteomics (iMPAQT)

**Dan Nakano[1], Takumi Kawaguchi**[1]*, **Hideki Iwamoto**[1,2], **Masako Hayakawa**[2], **Hironori Koga**[1,2], **Takuji Torimura**[1,2]

**1** Division of Gastroenterology, Department of Medicine, Kurume University School of Medicine, Kurume, Japan, **2** Liver Cancer Division, Research Center for Innovative Cancer Therapy, Kurume University, Kurume, Japan

* takumi@med.kurume-u.ac.jp

## Abstract

### Aim

Metabolic reprograming is crucial in the proliferation of hepatocellular carcinoma (HCC). Canagliflozin (CANA), a sodium-glucose cotransporter 2 (SGLT2) inhibitor, affects various metabolisms. We investigated the effects of CANA on proliferation and metabolic reprograming of HCC cell lines using multi-omics analysis of metabolomics and absolute quantification proteomics (iMPAQT).

### Methods

The cells were counted 72 hours after treatment with CANA (10 μM; n = 5) or dimethyl sulfoxide (control [CON]; n = 5) in Hep3B and Huh7 cells. In Hep3B cells, metabolomics and iMPAQT were used to evaluate the levels of metabolites and metabolic enzymes in the CANA and CON groups (each n = 5) 48 hours after treatment.

### Results

Seventy-two hours after treatment, the number of cells in the CANA group was significantly decreased compared to that in the CON group in Hep3B and Huh7 cells. On multi-omics analysis, there was a significant difference in the levels of 85 metabolites and 68 metabolic enzymes between the CANA and CON groups. For instance, CANA significantly downregulated ATP synthase F1 subunit alpha, a mitochondrial electron transport system protein (CON 297.28 ±20.63 vs. CANA 251.83±22.83 fmol/10 μg protein; P = 0.0183). CANA also significantly upregulated 3-hydroxybutyrate, a beta-oxidation metabolite (CON 530±14 vs. CANA 854±68 arbitrary units; P<0.001). Moreover, CANA significantly downregulated nucleoside diphosphate kinase 1 (CON 110.30±11.37 vs. CANA 89.14±8.39 fmol/10 μg protein; P = 0.0172).

**Data Availability Statement:** All relevant data are within the manuscript and its Supporting Information files.

**Funding:** This research was supported by Japan Agency for Medical Research and Development (AMED) under Grant Number JP19fk0210040.

**Competing interests:** Takumi Kawaguchi received lecture fees from Mitsubishi Tanabe Pharma Corporation, MSD K.K., and Otsuka Pharmaceutical Co., Ltd. However, this does not alter our adherence to PLOS ONE policies on sharing data and materials. The description was added in the revised main text as well as the cover letter.

**Abbreviations:** HCC, hepatocellular carcinoma; ATP, adenosine triphosphate; SGLT2, sodium-glucose cotransporter 2; CANA, canagliflozin; SGLT2i, SGLT2 inhibitor; iMPAQT, in vitro proteome-assisted multiple reaction monitoring for protein absolute quantification; GLUT, glucose transporter; AMPK, adenosine monophosphate-activated protein kinase; CPT, carnitine palmitoyl transferase; PBS, phosphate buffered saline; DMSO, dimethyl sulfoxide; CON, control; AN, annexin V; 7AAD, 7-amino-actinomycin D; PARP, Poly (adenosine diphosphate-ribose) polymerase; ACAA1, acetyl-Coenzyme A acyltransferase 1; UDP, uridine diphosphate; SCD, stearoyl- CoA desaturase; NME1, nucleoside diphosphate kinase 1; PRIM2, DNA primase, polypeptide 2.

## Conclusions

We found that CANA suppressed the proliferation of HCC cells through alterations in mitochondrial oxidative phosphorylation metabolism, fatty acid metabolism, and purine and pyrimidine metabolism. Thus, CANA may suppress the proliferation of HCC by regulating metabolic reprograming.

## Introduction

Hepatocellular carcinoma (HCC) is the second leading cause of cancer-related death worldwide [1]. Although there are several therapeutic options for HCC including oral multikinase inhibitors, the prognosis of patients with HCC is still unsatisfactory [1]. One mechanism of tumor progression and treatment resistance is metabolic reprograming, which promotes adenosine triphosphate (ATP) production to meet the bioenergetic and biosynthetic demands of tumor growth [2]. In HCC, metabolic reprograming is seen in various metabolisms including lipid, amino acid, and purine metabolisms [3–5]. In addition, reprograming of glucose metabolism is involved in the proliferation of HCC [6–8].

Recently, sodium glucose co-transporter 2 (SGLT2), a glucose transporter, has been found to occur not only in renal proximal tubular epithelial cells but also in cancer cells including pancreatic cancer as well as HCC [9]. In addition, a meta-analysis showed that canagliflozin (CANA), a SGLT2 inhibitor (SGLT2i), suppresses gastrointestinal cancers in patients with type 2 diabetes mellitus [10]. Kaji et al. demonstrated that CANA inhibits hepatoma cell growth by suppressing angiogenic activity and chronic inflammation [11]. Moreover, Shiba et al. reported that CANA attenuates the development of HCC by reducing the oxidative stress of adipose tissue in a mouse model of nonalcoholic steatohepatitis [12]. However, the direct effects of SGLT2i on metabolic reprograming in HCC remain unclear.

Metabolomics is the large-scale systematic analysis of metabolites, which is a powerful approach to uncovering detailed information about changes in metabolites [13]. Metabolomics has been applied to the study of HCC and provides new insights into the diagnosis, prognosis, and therapeutic evaluation of HCC [13, 14]. Changes in metabolites are caused by alterations in metabolic enzymes; therefore, proteomics should also be applied to detect metabolic reprograming. Recently, Matsumoto et al. developed a new method for in vitro proteome-assisted multiple reaction monitoring for protein absolute quantification (iMPAQT) [15]. iMPAQT is able to measure the absolute quantity of any human protein in a given pathway of interest with high quantitative accuracy [15]. Although iMPAQT has been used to investigate the therapeutic target of bladder cancer [16], this new proteomics analysis has never been applied to investigate metabolic reprogramming in HCC.

The aim of this study was to investigate the effects of SGLT2i on the proliferation of HCC cell lines. In addition, we investigated the effects of SGLT2i on metabolic reprograming using multi-omics analysis combining metabolomics and iMPAQT analyses.

## Methods

### Reagents and antibodies

CANA, a SGLT2i, was kindly provided by the Mitsubishi Tanabe Pharma Corporation (Osaka, Japan). All other reagents were purchased from Cell Signaling Technology, Inc. (Danvers, MA) unless otherwise indicated. Antibodies against SGLT2 were purchased from Abcam plc

(ab137207 and ab85626, Kenbridge, England), Proteintech Group, Inc. (24654-1-AP, Rosemont, IL), and Cell Signaling Technology, Inc. (#14210). Antibodies against glucose transporter (GLUT) 4, GLUT5, GLUT6, anti-mitochondrial pyruvate dehydrogenase kinase, hydroxymethylglutaryl-CoA lyase, and phosphate- AMP-activated protein kinase (AMPK) α2 (Ser345) were purchased from Abcam plc. Antibodies against GLUT1, GLUT2, GLUT3, caspase8, caspase3, carnitine palmitoyltransferase (CPT) 1A, CPT2, acyl-CoA synthetase long-chain family member/fatty acid-CoA ligase, and solute carrier family 25 member 2 were purchased from Proteintech Group, Inc. Actin antibody was purchased from Sigma-Aldrich Co. LLC (St. Louis, MO). Anti-glyceraldehyde-3-phosphate dehydrogenase antibody was purchased Santa Cruz Biotechnology (Dallas, TX). Jurkat (Human) whole cell lysate was purchased from Abcam plc. Primary human hepatocytes (LHum17003) was purchased from BIOPREDIC International (Saint-Grégoire, France).

## Cell lines

Huh7 cells were obtained from HuH7 (JCRB0403) and HLF (JCRB0405) cells were obtained from the JCRB Cell Bank (Tokyo, Japan). Hep3B cells (HB8064) were obtained from the American Type Culture Collection (ATCC-LGC Standards). HAK-1A, HAK-1B, KYN-2, and KMCH-1 were kindly provided from Prof. Hirohisa Yano (Department of Pathology, Kurume University School of Medicine, Kurume, Japan).

Eight hepatoma cell lines such as Huh7, HLF, HepG2, Hep3B, KYN2, KMCH1, HAK1A, and, HAK1B cells were maintained in modified Dulbecco's modified Eagle's medium with L-glutamine and phenol red (Wako, Osaka, Japan), penicillin (10,000 units/mL), and streptomycin (10 mg/mL) at 37°C in a humidified atmosphere contained 5% $CO^2$.

## Immunoblotting analysis and quantification

After washes with phosphate buffered saline (PBS), cells were lysed in a lysis buffer (RIPA buffer, Thermo Fisher Scientific, Inc., Waltham, MA) containing a protease and phosphatase inhibitor cocktail (Nacalai Tesque, Inc.). Cell lysates were centrifuged at 15,000 rpm × g for 15 minutes at 4°C, and the supernatant was collected. The protein concentration was determined by a protein assay kit (Pierce™, Thermo Fisher Scientific, Inc.). The samples were then mixed with an equal volume of 2 × sample loading buffer containing 4 × lithium dodecyl sulfate sample buffer (NuPAGE® LDS sample buffer, Thermo Fisher Scientific, Inc.) and 2-mercaptoethanol and a sample reducing agent (NuPAGE® Sample Reducing Agent, Thermo Fisher Scientific, Inc.). The membranes were washed and incubated with horseradish peroxidase-labeled secondary antibodies (GE Healthcare UK Ltd., Buckinghamshire, UK) for 1 hour at room temperature. After several washes, the membranes were incubated with chemiluminescent reagents (ImmunoStar LD, FUJIFILM Wako Pure Chemical Corporation, Tokyo, Japan), and specific bands were visualized by an image analyzer LAS-4000mini (GE Healthcare) as previously described [17]. The quantification of protein expression in western blotting was evaluated using ImageJ software (National Institutes of Health, Bethesda, MD, USA) [18].

## Immunofluorescence staining

In this immunofluorescence staining, Huh7 and Hep3B cells were used. Cells were fixed in 4% paraformaldehyde and were incubated at 37°C for 20 minutes. After washes with PBS, cells were treated with 0.1% TritonX (Sigma-Aldrich Co. LLC.) and incubated at 37°C for 5 minutes. Cells were incubated with 3% skim milk in 1 × PBS at room temperature for 20 minutes. Antibodies were diluted in 3% skim milk in PBS. Cells were incubated in primary antibodies for SGLT2 (ab85626, Abcam plc) at 4°C overnight, washed with PBS, and then incubated in fluorescent-dye conjugated secondary antibody at 4°C in the dark, overnight. The nucleus was

stained by 4',6-diamidino-2-phenylindole. The mitochondria were stained by an antibody for mitochondrial pyruvate dehydrogenase kinase 1. A fluorescence microscope (BZ-X700; Keyence Corporation, Osaka, Japan) was used to visualize the distribution of immunostaining for SGLT2 as previously described [19].

## Isolation of mitochondrial fraction

Mitochondrial fraction was isolated by using Mitochondria Isolation Kit for Cultured Cells (ab110171, Abcam) according to the manufactures' instruction. Briefly, Hep3B and Huh7 cells were collected with a cell lifter and pelleted by centrifugation at 1,000 g. The cells were freezed and then thaw in order to weaken the cell membranes. The cells were resuspended in Reagent A and, then, were transferred into a pre-cooled Dounce Homogenizer. The homogenates were centrifuged at 1,000 g for 10 minutes at 4°C and save as supernatant #1. The pellet was resuspended in Reagent B and repeated the rupturing. The homogenates were centrifuged and save as supernatant #2. Supernatants #1 and #2 were combined and centrifuged. The pellet was collected and resuspend into 500 μL of Reagent C supplemented with Protease Inhibitors. Freeze the aliquots at -80°C until use.

## Effect of CANA on cell proliferation

The Hep3B and Huh7 cells were counted at 0, 24, 48, and 72 hours after treatment with 3, 10, and 30 μM of CANA (n = 5 per condition) or dimethyl sulfoxide (DMSO) (control [CON]; n = 5). Cells were trypsinized after washing with PBS. Then, the number of cells was determined using an automated cell counter (CDA-500; Sysmex Corporation, Kobe, Japan) as previously described [20]. In addition, the Hep3B cells were counted at 0, 24, 48, and 72 hours after treatment with 10 μM of dapagliflozin.

## Living cell assessment by reagent SF assay

In this assay, Hep3B cells were used. The living Hep3B cells were counted by using Cell Count Reagent SF (Nacalai Tesque, Inc., Kyoto, Japan) according to the manufacturer's instructions. This assay is a sensitive colorimetric assay utilizing a highly water-soluble tetrazolium salt, which produces a water-soluble formazan dye upon reduction in the presence of an electron carrier. Briefly, cells were cultured in Dulbecco's modified Eagle medium with 10% fetal bovine serum at 37°C for 2 hours. Then, cells were treated with 3 μM, 10 μM of CANA, or DMSO (CON, each n = 5) and incubated at 37°C for 72 hours. Then, 10 μL of Cell Count Reagent SF was added to each dish, and the cells were incubated for 30 minutes. Then the absorbance was read at 490 nm with a plate reader (BZ-X700; Keyence Corporation), and the number of viable cells was determined using the absorbance value of a previously prepared calibration curve.

## Dead cell assessment by trypan blue exclusion assay

In this assay, Hep3B cells were used. Cells were cultured in Dulbecco's modified Eagle's medium with 10% fetal bovine serum at 37°C for 2 hours. Then, cells were treated with 10 μM, 30 μM of CANA, DMSO, or mitomycin (positive control for apoptosis) (each n = 5) and incubated at 37°C for 72 hours. After PBS washes, cells were treated with 1 mL of trypsin ethylenediaminetetraacetic acid (Nacalai Tesque, Inc.) and incubated at 37°C for 5 minutes, and medium was added to make a total of 3 mL. Ten μL of these samples was mixed with 10 μL of GIBCO Trypan Blue Stain, 0.4% (Thermo Fisher Scientific, Inc.). Then, the total and dead cells were counted by BIO RAD TC20 Automated Cell Counter (Bio-Rad Laboratories, Inc. Hercules, CA) as previously described [21].

## Evaluation of morphological change of Hep3B and Huh7 cells

Hep3B and Huh7 cells were cultured with in Dulbecco's modified Eagle's medium containing10% fetal bovine serum, and cell morphology was observed by using a phase-contrast microscopy (BZ-X700; Keyence Corporation, Osaka, Japan) 48hours after treated with 3 μM, 10 μM of CANA, or DMSO.

## Evaluation of apoptosis

In this assay, Hep3B and Huh7 cells were used. Apoptosis was evaluated by using annexin V (AN)/7-amino-actinomycin D (7AAD) as previously described [22]. Briefly, Hep3B cells were cultured in Dulbecco's modified Eagle medium with 10% fetal bovine serum at 37˚C for 2 hours. Then, the cells were treated with CANA (3 μM), DMSO (CON), and mitomycin (positive control for apoptosis) (each n = 3) and incubated at 37˚C for 72 hours. The cell suspensions were washed and stained with fluorescein isothiocyanate-labeled AN and 7AAD. A fluorescence microscope (BZ-X700; Keyence Corporation) was used to visualize the distribution of immunostaining for AN and 7AAD. The cells were analyzed using flow cytometry (EPICS profile, Coulter, Hialeah, FL), and the AN−/7AAD−, AN+/7AAD− and AN+/7AAD + populations, which have been found to correspond to live cells, early apoptotic cells, and both late apoptotic and necrotic cells, respectively, were counted.

Apoptosis was evaluated by immunoblotting in Hep3B and Huh7 cells using antibodies against caspase8, caspase3, and Poly (adenosine diphosphate-ribose) polymerase (PARP).

## Cell cycle analysis

In this assay, Huh7 and Hep3B cells were used. After treatment with DMSO or 10 μM of CANA at 37˚C for 72 hours, the cells were trypsinized. The cells were fixed in 75% ethanol and PBS for 5 minutes at -20˚C and centrifuged at 2,000 rpm × g for 10 minutes at 4˚C. The samples were incubated with PBS including RNase and propidium iodide for DNA staining at 37˚C for 5 minutes. The DNA content was assessed by monitoring with FACSCalibur (Becton Dickinson, Franklin Lakes, NJ). The flow cytometry data were collected, and cell cycle distributions were analyzed with Cell Quest software (Becton Dickinson) as previously described [23].

## Metabolomic analysis

In this assay, Hep3B cells were used. Cell lysate samples 48 hours after treatment with DMSO (CON) and CANA (10 μM) were used for metabolomic analysis (each n = 5). Metabolome measurements were performed at a service facility of LSI Medience Corporation (Tokyo, Japan) as previously described [24]. Briefly, cell lysate was added to methanol and then mixed for 15 minutes with a shaker at room temperature. After centrifugation at 10,000 g for 10 minutes, the supernatant was dried with nitrogen gas, and the residue was dissolved with 10% acetonitrile aqueous solution. After adding internal standards, they were analyzed with both liquid chromatography mass spectrometry and capillary electrophoresis coupled with mass spectrometry. Tuning and calibration were performed with a standard solution provided by Agilent Technology, and the resolution errors were controlled within 3 ppm. The order of measurement was randomized to minimize the specific error in each group. Quality control samples were prepared by pooling samples.

## iMPAQT

In this study, we employed iMPAQT in order to perform a global analysis for absolute quantification of protein expression simultaneously in Hep3B cell both CANA (10 μM) and CON

group. The analysis was performed as previously described [15]. Briefly, cells ($2 \times 10^6$) were lysed with 150 μL of lysis buffer (a solution containing 2% SDS, 7 M urea, and 100 mM Tris-HCl, pH 8.8), and the samples were diluted with an equal volume of water. The protein concentrations of the samples were determined with BCA assays (Bio-Rad Laboratories, Inc., Hercules, CA). To block cysteine/cysteine residues, we treated 200 μg of each sample with 5.0 mM Tris (2-carboxyethyl) phosphine hydrochloride (Thermo Fisher Scientific, Inc.) for 30 minutes at 37˚C, then alkylation was performed with 10 mM 2-iodoacetoamide (Sigma-Aldrich Co., LLC., St. Louis, MO) for 30 minutes at room temperature. After these samples were subjected to acetone precipitation, the resulting pellet was resuspended in 100 μL digestion buffer (0.5 M triethylammonium bicarbonate). Each sample was digested with lysyl-endopeptidase (2 μg, Wako) for 3 hours at 37˚C. Subsequently, the samples were further digested with trypsin (4 μg, Thermo Fisher Scientific, Inc.) for 14 hours at 37˚C. The resulting cell digests were freeze-dried and then labeled with the mTRAQ® Δ0 (light) reagent (SCIEX Co. Ltd., Ontario, Canada). Each sample was spiked with synthetic peptides (Funakoshi Co. Ltd., Tokyo, Japan) for internal standard, which treated reductive alkylation and mTRAQ® Δ4 (heavy) labeling (SCIEX Co., Ltd.). The samples were subjected to reverse-phase liquid chromatography followed by multiple reaction monitoring analysis. Experiments using mass spectrometry and pretreatment were performed by the service provider Kyusyu Pro Search LLP (Fukuoka, Japan).

## Statistical analysis

All data are expressed as the mean ± SD. Comparisons between any two groups were performed using the Wilcoxon rank-sum test. Comparisons among multiple groups were made using one-way analyses of variance, followed by Fisher's protected least-significant-difference post-hoc test. These statistical analysis were performed using JMP Pro® 13 (SAS Institute Inc., Cary, NC). A P value <0.05 was considered statistically significant.

## Results

### Expression of SGLT2

In immunoblotting, expression of SGLT2 was seen in Hep3B and Huh7 cells, and the expression was confirmed by 4 different antibodies against SGLT2 (Fig 1A). We also investigated expression of SGLT2 in 8 hepatoma cell lines such as Huh7, HLF, HepG2, Hep3B, KYN2, KMCH1, HAK1A, and, HAK1B cell lines by western blotting. SGLT2 occurred in the 8 hepatoma cell lines (S1 Fig). In addition, we investigated expression of SGLT2 in Jurkat cells as a SGLT2-positive sample and primary human hepatocytes. Expression of SGLT2 was seen in the both cells, although the expression was weak in primary human hepatocytes.

In immunostaining, expression of SGLT2 was seen in the cytoplasm of Hep3B and Huh7 cells (Fig 1B). SGLT2 co-localized with mitochondrial pyruvate dehydrogenase kinase 1 in Hep3B and Huh7 cells in double immunostaining (Fig 1C). Expression of SGLT2 was also detected in the mitochondrial fraction of Hep3B and Huh7 cells. On the other hand, expression of SGLT2 was weak in cytoplasmic fraction of Hep3B and Huh7 cells (Fig 1D). Expression of SGLT2 was not detected in the negative control examination, which lacks primary antibody for SGLT2 (Fig 1D) In immunoblotting, expression of SGLT1 and GLUT1, 2, 3, 5, and 6 was seen in Hep3B and Huh7 cells (S2 Fig).

### Effects of CANA on proliferation of Hep3B and Huh7 cells

In Hep3B cells, the number of cells was significantly decreased in the 10 μM and 30 μM CANA group compared to the CON group, and this decreased in a dose-dependent fashion (Fig 2A).

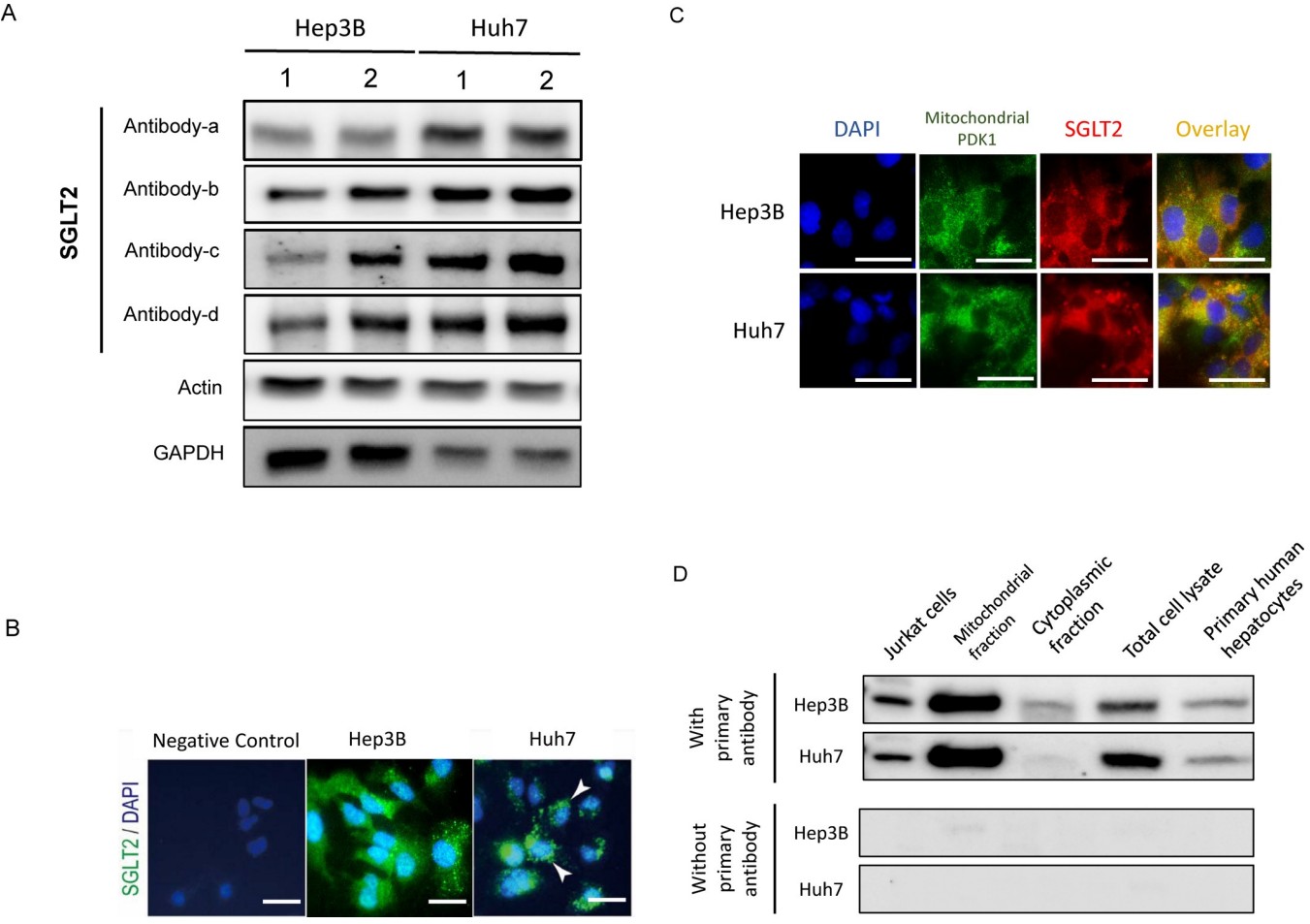

**Fig 1. Expression of SGLT2 in Hep3B and Huh7.** A) Immunoblotting using 4 different antibodies for SGLT2: Antibody-a, Proteintech Group, Inc. (24654-1-AP); Antibody-b, Abcam plc. (ab137207); Antibody-c, Abcam plc. (ab85626), Antibody-d, Cell Signaling Technology, Inc. (#14210). B) Immunohistochemistry for SGLT2. Green indicates SGLT2. The nucleus was stained by DAPI. Scale bar = 50 μm. C) Double immunostaining for mitochondrial PDK-1 and SGLT2. Green indicates mitochondrial PDK-1. Red indicates SGLT2. Yellow indicates colocalization of mitochondrial PDK-1 and SGLT2. The nucleus was stained by DAPI. Scale bar = 50 μm. D) Immunoblotting for SGLT2 in mitochondrial and cytoplasmic fractions. Abbreviations: SGLT2, sodium-glucose cotransporter 2; GAPDH, glyceraldehyde-3-phosphate dehydrogenase; DAPI, 4',6-diamidino-2-phenylindole; PDK-1, pyruvate dehydrogenase kinase 1.

Similar findings were also seen in Huh7 cells (Fig 2B). On the other hand, there was no significant difference in cell number between 10 μM dapagliflozin group and CON group in Hep3B cells (S3 Fig).

In the living cell count assay using SF, the live cell rate of Hep3B was significantly decreased in the 10 μM and 30 μM CANA groups compared to the CON group (Fig 2C). In trypan blue staining, the number of dead Hep3B cells was significantly higher in the 30 μM CANA group than in the CON group. However, there was no significant difference in the number of dead cells between the 10 μM CANA and CON groups (Fig 2D).

## Effects of CANA on morphological change and apoptosis of Hep3B and Huh7 cells

The impact of CANA on morphological change of Hep3B and Huh7 cells was examined by using phase-contrast-microscope 48 hours after treatment. There was no morphological

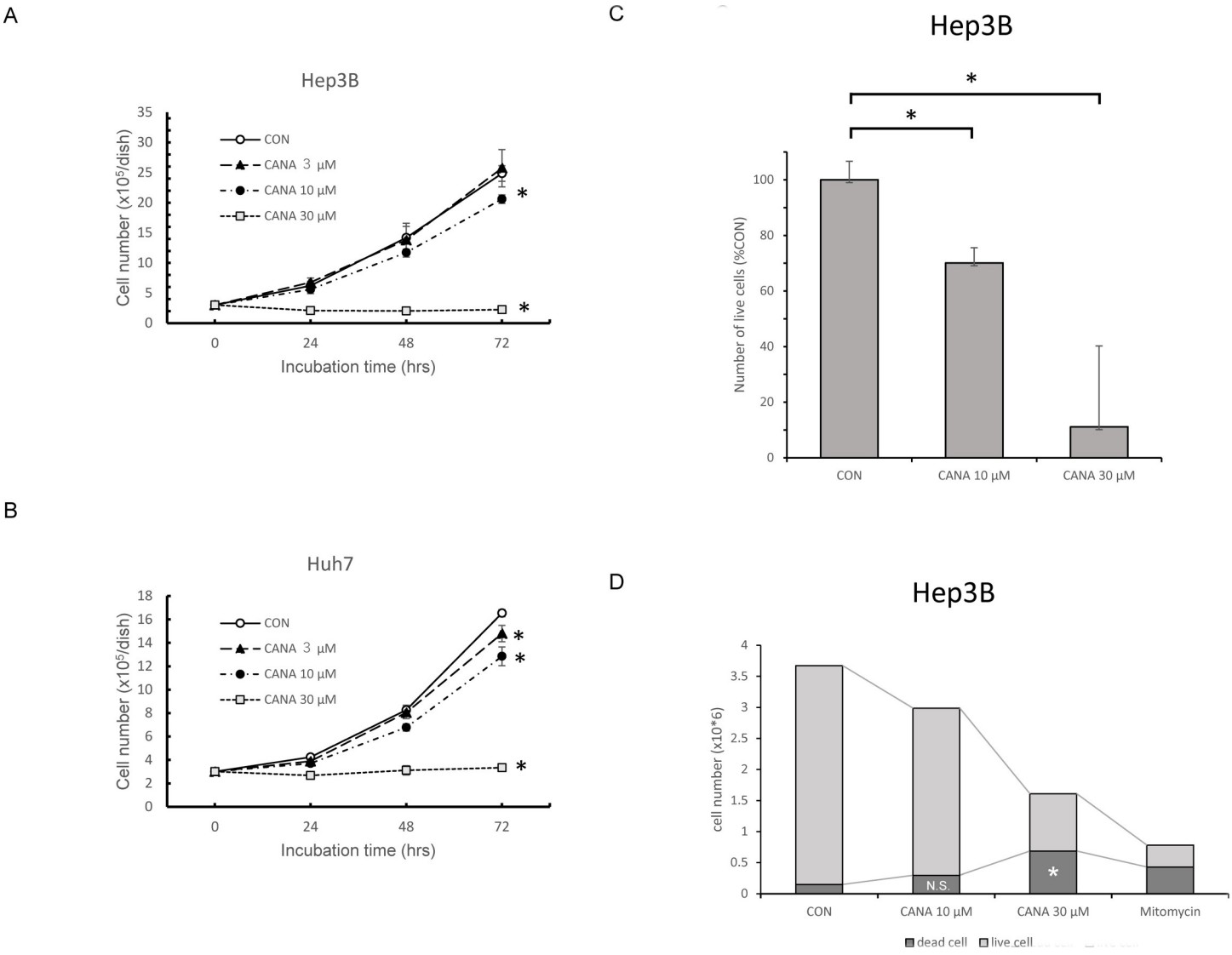

**Fig 2.** Effects of CANA on proliferation of A) Hep3B and B) Huh7 cells. C) Effects of CANA on live cell rate of Hep3B. D) Effects of CANA on the number of dead Hep3B cells. * P<0.01. Abbreviations: CON, control; CANA, canagliflozin.

change in Hep3B and Huh7 cells after 10 μM of CANA treatment (S4 Fig). However, after treatment with 30 μM CANA, there were morphological changes such as spindled and/or rounded shapes in Hep3B and Huh7 cells (S4 Fig).

The impact of apoptosis on the CANA-caused decrease in the number of cells was evaluated by using AN and 7AAD in Hep3B cells. In flow cytometry, the percentages of necrotic cells were 12.1% in the Mitomycin group (Fig 3A; positive control). However, the percentage was 2.03% and 2.34% in the CON and 10 μM CANA groups, respectively (Fig 3B and 3C). There was no significant difference in the percentages of necrotic cells between the CON and 10 μM CANA groups (n = 4; CON 1.90±0.22% vs. CANA 2.59±0.22%, P = 0.067). In immunostaining, the expression of AN and 7AAD was up-regulated in the Mitomycin group (Fig 3D). Meanwhile, the expression of AN and 7AAD was weak in the 10 μM CANA and CON groups (Fig 3E and 3F). In the Mitomycin group, there was an upregulation in p18 cleaved caspase 8, p17 cleaved caspase 3, and cleaved PARP in Hep3B cells. (S5A Fig). Meanwhile, there was no

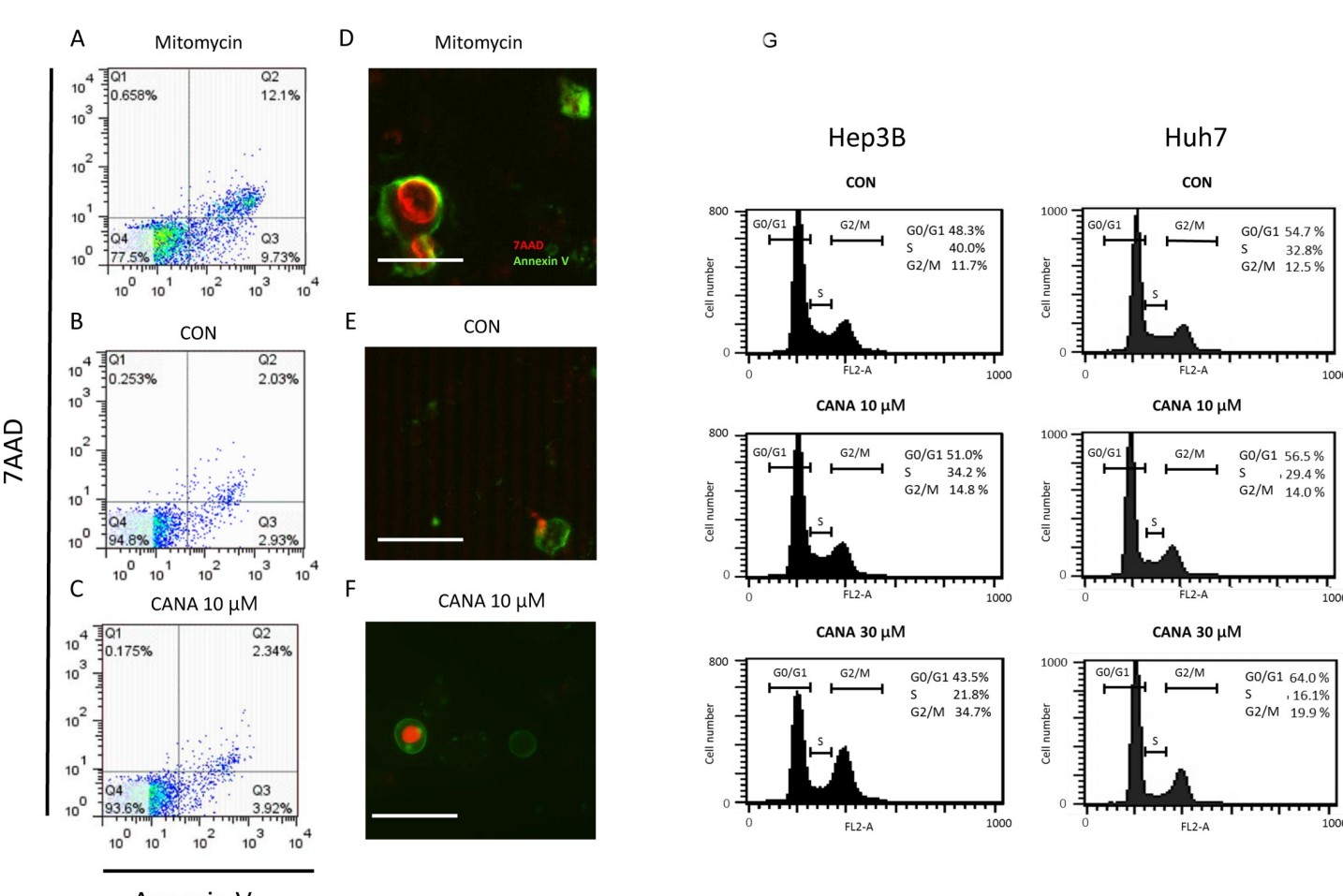

**Fig 3. Effects of CANA on apoptosis and cell cycle of Hep3B and Huh7 cells.** Apoptosis was evaluated by flow cytometry using 7AAD and annexin V in the A) Mitomycin group, B) CON group, and C) 10 μM CANA groups. Apoptosis was also evaluated by immunostaining for 7AAD (red) and annexin V (green) in the D) Mitomycin group, E) CON group, and F) 10 μM CANA group. Scale bar = 50 μm. G) Cell cycle distribution was evaluated by flow cytometry. Abbreviations: 7AAD, 7-amino-actinomycin D; CON, control; CANA, canagliflozin.

significant difference in the expression of p18 cleaved caspase 8, p17 cleaved caspase 3, and cleaved PARP between the 10 μM CANA and CON groups in Hep3B cells (S5A Fig). Similarly, there was no significant difference in the expression of apoptosis-related molecules including cleaved PARP between the CANA and CON groups in Huh7 cells. (S5B Fig). By treatment with 30 μM CANA, there was an upregulation of cleaved PARP in Hep3B and Huh7 cells.

## Effects of CANA on cell cycle in Hep3B and Huh7 cells

At 72 hours after treatment, the percentage of cells in the G2/M phase was 11.7±0.4% in the CON group. Meanwhile, in Hep3B cells, the percentage was significantly increased to 14.8 ±0.1% and 34.7±0.5% in the 10 μM and 30 μM CANA groups (both P<0.01), respectively (Fig 3G). Similarly, in Huh7 cells, the percentage of G2/M phase was significantly increased with dose-dependent fashion (12.5±0.2% vs. 14.0±0.2% and 19.9±0.7%; CON vs. 10 μM and 30 μM CANA groups; both P<0.01), respectively (Fig 3G).

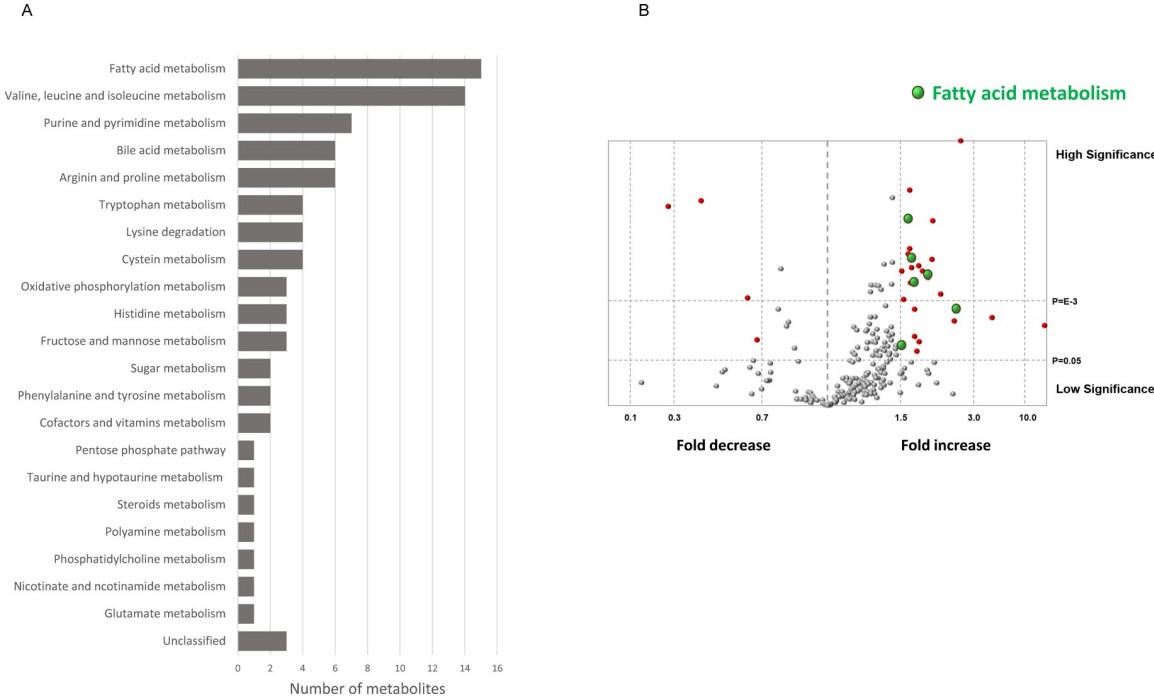

**Fig 4. Effects of CANA on metabolites evaluated by metabolomics.** A) The number of significantly altered metabolites between the CANA and CON groups was plotted according to each category of metabolism. B) Volcano plot of differential metabolomics between the CON and CANA groups. Red circle indicates differential metabolites, which determined the condition of >1.5-fold increase with P<0.05 or <0.7-fold decrease with P<0.05. The green circle indicates differential metabolites associated with fatty acid metabolism. The gray circle indicates non-differential metabolites, which determined the condition of ≤1.5-fold increase and ≥0.7-fold decrease with P<0.05 and non-differential metabolites with P≥0.05. Abbreviations: CON, control; CANA, canagliflozin.

## Effects of CANA on metabolites evaluated by metabolomics

In a metabolomic analysis, the effects of CANA on 225 metabolite levels were evaluated (S1 Table). A significant difference was seen between the CON and CANA groups in 85 metabolites (Fig 4A). Of these, the most altered metabolism was fatty acid metabolism (15 metabolites), followed by valine leucine and isoleucine metabolism (14 metabolites), then purine and pyrimidine metabolism (7 metabolites) (Fig 4A).

The differences in 225 metabolites between the CON and CANA groups are demonstrated by a volcano plot, and significantly altered metabolites are highlighted in red (Fig 5B). The volcano plot shows that metabolites associated with fatty acid elongation such as acetylcarnitine, butyrylcarnicine, and 3-hydroxybutanoate were significantly up-regulated in the CANA group compared to the CON group (green in Fig 5B). In addition, metabolites associated with fatty acid biosynthesis such as erucic acid and myristoleate were significantly up-regulated in the CANA group (green in Fig 5B). Thus, fatty acid metabolism was significantly altered in the CANA group.

## Effects of CANA on expression level of metabolic enzymes by iMPAQT

In iMPAQT, the effects of CANA on the expression levels of 342 metabolic enzymes were evaluated. A significant difference was seen in 68 metabolic enzymes between the CON and CANA groups (S2 Table). Of these, the most altered metabolism was oxidative phosphorylation (11 metabolic enzymes), followed by purine and pyrimidine metabolism (7 metabolic enzymes). In addition, 4 and 3 metabolic enzymes were categorized as fatty acid metabolism and valine leucine and isoleucine metabolism, respectively (Table 1).

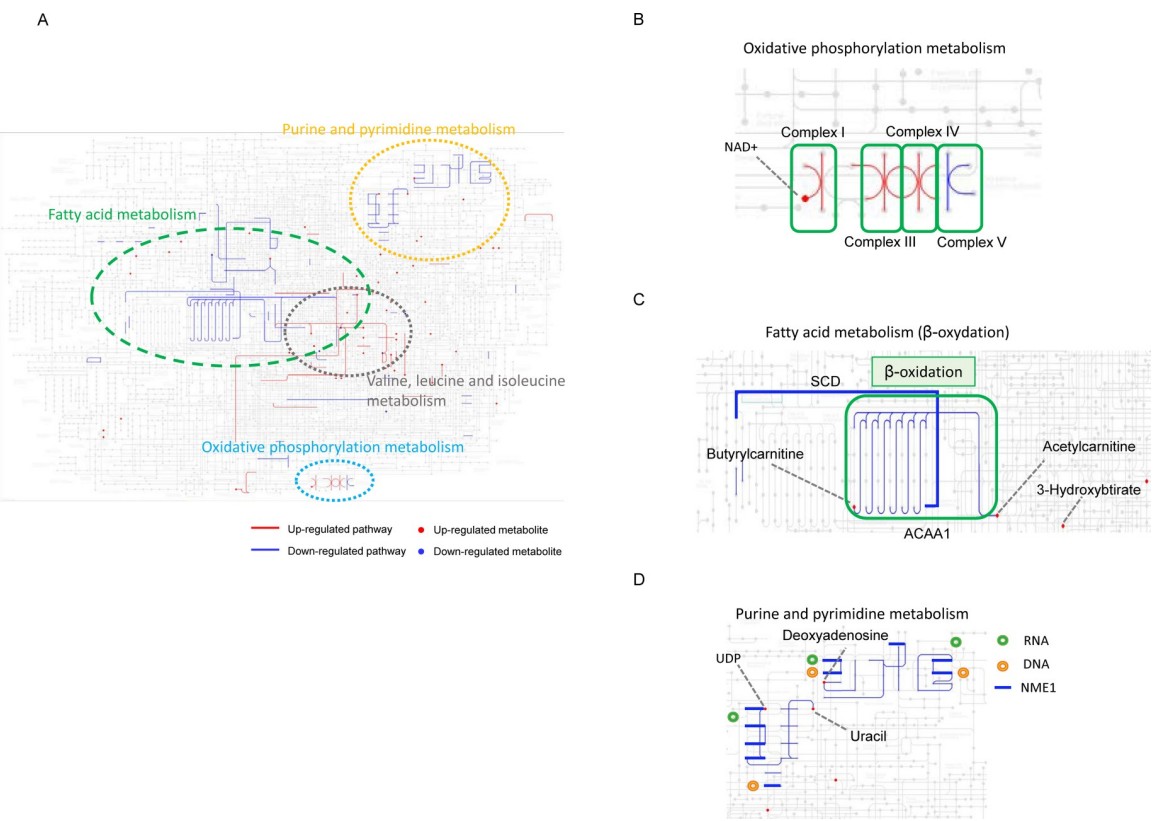

**Fig 5. Metabolism map showing both CANA-altered metabolites and metabolic enzymes using multi-omics analysis of metabolomics and iMPAQT.** Red line indicates an up-regulated pathway. Red circle indicates an up-regulated metabolite. Blue line indicates a down-regulated pathway. Blue circle indicates a down-regulated metabolite. A) Whole metabolism map. B) Map for oxidative phosphorylation metabolism. C) Map for fatty acid metabolism. Thick blue line indicates SCD pathway. Green rectangle indicates ACAA1 pathways. D) Map for purine and pyrimidine metabolism. Thick blue line indicates NME1 pathways. Abbreviations: NAD+, nicotinamide adenine dinucleotide; SCD, stearoyl-CoA desaturase; ACAA1, acetyl-coenzyme A acyltransferase 1; UDP, uridine diphosphate; RNA, ribonucleic acid; DNA, deoxyribonucleic acid; NME1, nucleoside diphosphate kinase1.

## Metabolism map showing both CANA-altered metabolites and metabolic enzymes using multi-omics analysis of metabolomics and iMPAQT

In the metabolite map, 85 metabolites and 68 metabolic enzymes altered by CANA were plotted (Fig 5A). We found that the altered metabolites and metabolic enzymes could be classified into the following 4 categories; 1) oxidative phosphorylation metabolism, 2) fatty acid metabolism, 3) purine and pyrimidine metabolism, and 4) valine, leucine, and isoleucine metabolism (Fig 5A).

In oxidative phosphorylation metabolism, CANA significantly upregulated NAD$^+$ in metabolomics (red circle in Fig 5B). Although CANA significantly upregulated 10 proteins associated with the electron transport system, including cytochrome c oxidase subunit 7A2, CANA downregulated a protein associated with the electron transport system, including ATP synthase F1 subunit alpha in iMPAQT (blue line in Fig 5B).

In fatty acid metabolism, CANA significantly upregulated 4 metabolites associated with beta-oxidation including butyrylcarnitine, acetylcarnitine, and 3-hydroxybutyrate in metabolomics. Meanwhile, in iMPAQT, CANA significantly downregulated acetyl-coenzyme A acyltransferase 1 (ACAA1), which is a key enzyme regulating beta-oxidation and production of ketone bodies including 3-hydroxybutyrate. CANA also significantly downregulated stearoyl-

**Table 1. Effects of CANA on expression level of metabolic enzymes by iMPAQT assay.**

| Enzyme | Pathway | Control | | SGLT2i | | P |
|---|---|---|---|---|---|---|
| | | Mean | SD | Mean | SD | |
| UQCRQ | Oxidative phosphorylation | 28.99 | 1.38 | 35.53 | 1.39 | 0.0002 |
| NDUFS2 | Oxidative phosphorylation | 34.84 | 0.77 | 39.48 | 1.26 | 0.0002 |
| NDUFB7 | Oxidative phosphorylation | 25.08 | 2.96 | 34.58 | 3.36 | 0.0028 |
| NDUFA9 | Oxidative phosphorylation | 41.70 | 2.44 | 49.16 | 2.62 | 0.0031 |
| COX7A2 | Oxidative phosphorylation | 83.33 | 10.93 | 111.67 | 9.13 | 0.0041 |
| NDUFV1 | Oxidative phosphorylation | 19.95 | 0.92 | 23.63 | 1.75 | 0.0059 |
| NDUFV2 | Oxidative phosphorylation | 34.97 | 1.03 | 39.36 | 2.17 | 0.0065 |
| UQCRC2 | Oxidative phosphorylation | 51.12 | 4.56 | 64.31 | 5.91 | 0.0077 |
| NDUFB4 | Oxidative phosphorylation | 25.42 | 1.21 | 28.65 | 1.58 | 0.0118 |
| ATP5A1 | Oxidative phosphorylation | 297.28 | 20.63 | 251.83 | 22.83 | 0.0183 |
| COX4I1 | Oxidative phosphorylation | 136.81 | 5.26 | 150.79 | 8.03 | 0.0195 |
| PRIM2 | Purine, Pyrimidine metabolism | 5.99 | 0.14 | 5.09 | 0.26 | 0.0003 |
| RRM1 | Purine, Pyrimidine metabolism | 23.03 | 0.71 | 17.31 | 2.20 | 0.0011 |
| ITPA | Purine, Pyrimidine metabolism | 27.46 | 2.17 | 22.17 | 1.31 | 0.0031 |
| ADSS | Purine, Pyrimidine metabolism | 26.76 | 0.86 | 22.79 | 2.18 | 0.0095 |
| NME1 | Purine, Pyrimidine metabolism | 110.30 | 11.37 | 89.14 | 8.39 | 0.0172 |
| NT5C2 | Purine, Pyrimidine metabolism | 21.82 | 1.06 | 19.31 | 1.55 | 0.0281 |
| PNP | Purine, Pyrimidine metabolism | 14.70 | 1.89 | 12.23 | 0.72 | 0.0408 |
| GPT2 | Arginine and proline metabolism | 10.41 | 1.32 | 19.14 | 2.32 | 0.0002 |
| GLUD1 | Arginine and proline metabolism | 167.21 | 7.59 | 129.09 | 9.27 | 0.0002 |
| PYCR1 | Arginine and proline metabolism | 8.82 | 0.87 | 11.86 | 1.35 | 0.0053 |
| PYCR2 | Arginine and proline metabolism | 30.88 | 3.28 | 36.72 | 1.49 | 0.0119 |
| CKB | Arginine and proline metabolism | 65.96 | 8.75 | 52.52 | 5.72 | 0.0331 |
| OAT | Arginine and proline metabolism | 16.32 | 1.10 | 18.79 | 1.63 | 0.0359 |
| SCD | Fatty acid metabolism | 90.82 | 5.18 | 68.71 | 5.07 | 0.0003 |
| ACOT7 | Fatty acid metabolism | 19.96 | 1.07 | 15.44 | 1.52 | 0.0012 |
| ACAT2 | Fatty acid metabolism | 55.37 | 3.71 | 40.13 | 6.47 | 0.0035 |
| ACAA1 | Fatty acid metabolism | 35.46 | 2.40 | 30.44 | 2.53 | 0.0206 |
| PHGDH | Glycine, serine and threonine metabolism | 41.67 | 2.53 | 67.88 | 6.00 | <0.0001 |
| SHMT2 | Glycine, serine and threonine metabolism | 155.76 | 4.86 | 194.10 | 13.99 | 0.0008 |
| PSAT1 | Glycine, serine and threonine metabolism | 109.65 | 12.76 | 170.17 | 24.36 | 0.0023 |
| CBS | Glycine, serine and threonine metabolism | 59.75 | 7.89 | 72.84 | 6.67 | 0.0350 |
| PKM2 | Glycolysis / Gluconeogenesis | 224.14 | 13.69 | 190.53 | 17.28 | 0.0159 |
| LDHA | Glycolysis / Gluconeogenesis | 197.54 | 11.83 | 163.32 | 19.95 | 0.0184 |
| ENO1 | Glycolysis / Gluconeogenesis | 614.47 | 41.47 | 509.49 | 62.87 | 0.0236 |
| GAPDH | Glycolysis / Gluconeogenesis | 1642.62 | 161.94 | 1369.61 | 130.00 | 0.0302 |
| PRPS2 | Pentose phosphate pathway | 18.68 | 1.91 | 15.36 | 1.20 | 0.0188 |
| ALDOA | Pentose phosphate pathway | 589.18 | 87.37 | 460.40 | 41.25 | 0.0285 |
| PGM1 | Pentose phosphate pathway | 8.18 | 0.59 | 6.95 | 0.85 | 0.0437 |
| GPI | Pentose phosphate pathway | 141.05 | 15.93 | 118.88 | 10.23 | 0.0473 |
| BCAT1 | Valine, leucine and isoleucine metabolism | 8.62 | 0.58 | 12.04 | 1.55 | 0.0033 |
| MUT | Valine, leucine and isoleucine metabolism | 8.83 | 0.60 | 10.54 | 0.69 | 0.0055 |
| IVD | Valine, leucine and isoleucine metabolism | 42.20 | 1.69 | 47.59 | 4.10 | 0.0411 |
| LPGAT1 | Glycerophospholipid metabolism | 47.08 | 4.29 | 40.23 | 3.47 | 0.0381 |
| LPCAT1 | Glycerophospholipid metabolism | 10.58 | 0.96 | 9.26 | 0.54 | 0.0433 |
| HMOX1 | Porphyrin and chlorophyll metabolism | 20.73 | 1.31 | 15.94 | 1.82 | 0.0027 |

*(Continued)*

**Table 1.** (Continued)

| Enzyme | Pathway | Control | | SGLT2i | | P |
|---|---|---|---|---|---|---|
| | | Mean | SD | Mean | SD | |
| HMOX2 | Porphyrin and chlorophyll metabolism | 19.50 | 0.42 | 17.93 | 1.16 | 0.0344 |
| TXNRD1 | Pyrimidine metabolism | 21.63 | 1.25 | 33.04 | 2.47 | <0.0001 |
| CTPS | Pyrimidine metabolism | 13.09 | 1.07 | 10.72 | 1.46 | 0.0310 |
| ME1 | Pyruvate metabolism | 22.97 | 2.89 | 28.00 | 1.70 | 0.0171 |
| GLO1 | Pyruvate metabolism | 37.90 | 7.06 | 26.46 | 3.08 | 0.0179 |
| LIPA | Steroid biosynthesis | 7.57 | 0.76 | 5.42 | 0.74 | 0.0037 |
| FDFT1 | Steroid biosynthesis | 9.71 | 1.13 | 7.34 | 1.05 | 0.0153 |
| ASNS | Alanine, aspartate and glutamate metabolism | 40.79 | 3.82 | 65.41 | 5.02 | 0.0001 |
| CMAS | Amino sugar and nucleotide sugar metabolism | 37.53 | 3.21 | 30.06 | 2.23 | 0.0051 |
| WARS | Aminoacyl-tRNA biosynthesis | 23.95 | 2.80 | 31.26 | 4.54 | 0.0253 |
| MAT2A | Cysteine and methionine metabolism | 51.83 | 4.87 | 39.93 | 4.37 | 0.0066 |
| PAFAH1B3 | Ether lipid metabolism | 23.57 | 3.71 | 19.03 | 1.23 | 0.0485 |
| GGH | Folate biosynthesis | 19.69 | 1.47 | 15.73 | 2.12 | 0.0155 |
| GMPPA | Fructose and mannose metabolism | 13.11 | 1.29 | 10.85 | 0.69 | 0.0149 |
| IMPA1 | Inositol phosphate metabolism | 39.71 | 2.13 | 35.59 | 2.52 | 0.0373 |
| AASDHPPT | Pantothenate and CoA biosynthesis | 19.10 | 1.90 | 14.65 | 1.19 | 0.0041 |
| DCXR | Pentose and glucuronate interconversions | 33.75 | 2.17 | 40.95 | 1.17 | 0.0004 |
| AGL | Starch and sucrose metabolism | 3.18 | 0.30 | 2.11 | 0.43 | 0.0032 |
| TST | Sulfur metabolism | 30.86 | 3.36 | 24.81 | 2.70 | 0.0231 |
| HMGCS1 | Synthesis and degradation of ketone bodies | 40.51 | 4.36 | 22.77 | 3.58 | 0.0002 |
| CAT | Tryptophan metabolism | 76.11 | 4.42 | 61.09 | 10.95 | 0.0345 |

CoA desaturase (SCD), which is implicated in the regulation of cell growth and differentiation (Fig 5C).

In purine metabolism, CANA significantly upregulated deoxyadenosine, a deoxyribonucleoside, in metabolomics. In addition, CANA significantly downregulated purine nucleoside phosphorylase, which is an enzyme of the nucleotide salvage pathways that produces nucleotide monophosphates (Fig 5D).

In pyrimidine metabolism, CANA significantly upregulated uridine diphosphate (UDP) and uracil on metabolomics. Moreover, in iMPAQT, CANA significantly downregulated RNA recognition motif 1 and nucleoside diphosphate kinase 1 (NME1), which are involved in the synthesis of nucleoside triphosphates, such as guanosine triphosphate, cytidine triphosphate, and uridine triphosphate (Fig 5D). Moreover, CANA significantly downregulated expression of DNA primase, polypeptide 2 (PRIM2), encoding a subunit of primase involved in purine and pyrimidine metabolism, DNA replication, and transcription (Table 1).

In valine, leucine, and isoleucine metabolism, CANA significantly upregulated valine, leucine, threonine, serine, and alanine in metabolomics. On the other hand, CANA downregulated BCAT1 in iMPAQT (S7 Fig).

## Effect of CANA on fatty acid metabolism-associated molecules

In Hep3B, there was no marked difference in the protein expression level of CPT1A, CPT2, acyl-CoA synthetase long-chain family member/fatty acid-CoA ligase, solute carrier family 25 member 2, hydroxymethylglutaryl-CoA lyase, fatty acid synthase, and CD36 between the CON and CANA groups (Fig 6). There was no significant difference in the protein expression level of AMPK between the two groups; however, p-AMPKα1 was significantly upregulated and p-

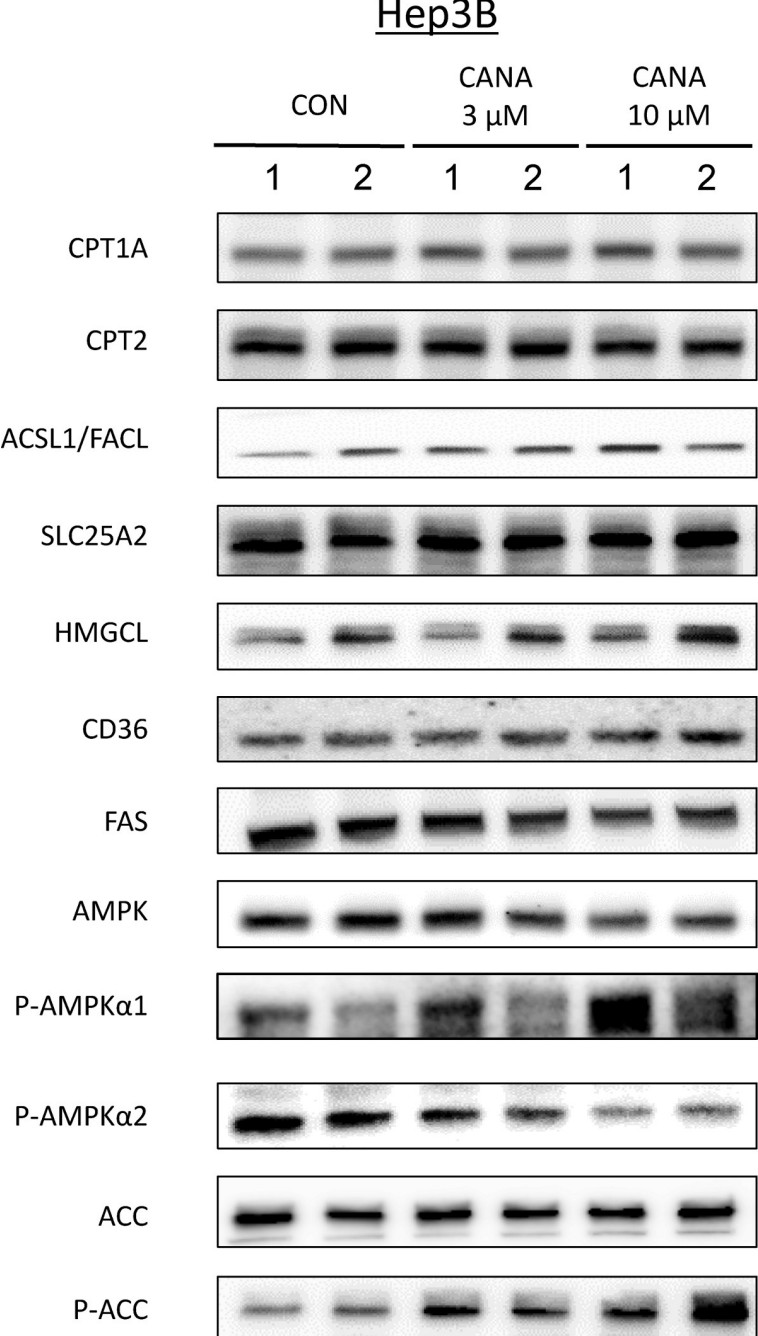

**Fig 6. Effect of CANA on fatty acid metabolism-associated molecules.** Expression of fatty acid metabolism-associated molecules was evaluated by immunoblotting 48 hours after treatment with CON, 3 μM CANA, and 10 μM CANA. Abbreviations: CPT, carnitine palmitoyltransferase; ACSL/FACL, acyl-CoA synthetase long-chain family member/fatty acid-CoA ligase; SLC25A2, solute carrier family 25 member 2, hydroxymethylglutaryl-CoA lyase; HMGCL, hydroxymethylglutaryl-CoA lyase; FAS, fatty acid synthase; AMPK, AMP-activated protein kinase; ACC, acetyl-CoA carboxylase.

AMPKα2 was significantly downregulated in the CANA 10 μM group compared to the CON group. No significant difference was seen in the protein expression level of ACC between the two groups; however, p-ACC was significantly upregulated in the CANA 10 μM group

compared to the CON group (Figs 6 and S8). Moreover, in Huh7 cells, we examined changes in phosphorylation of AMPKα1 and ACC. We found that CANA also phosphorylated AMPKα1 and ACC in Huh7 cells (S6 Fig).

## Discussion

In this study, SGLT2 occurred and localized on mitochondria in Hep3B and Huh7 cells. CANA significantly suppressed proliferation of these HCC cell lines. Multi-omics analysis of metabolomics and iMPAQT revealed that CANA mainly altered the following metabolisms; 1) oxidative phosphorylation metabolism, 2) fatty acid metabolism, and 3) purine and pyrimidine metabolism. Moreover, CANA altered phosphorylation of AMPK and ACC, which are sensors of intracellular ATP levels and regulators for beta oxidation. Thus, CANA may suppress proliferation of HCC cell lines via regulation of electron transport systems, beta oxidation, and nucleic acid synthesis.

We demonstrated that SGLT2 occurred in Hep3B and Huh7 cells. In normal tissue, SGLT2 occurs in the renal proximal tubules [25]. Moreover, SGLT2 is known to occur in various cancer cells including several HCC cell lines such as HepG2, Huh7, and JHH7 [26, 27]. Thus, our results were in good agreement with those of previous reports. In normal tissue, SGLT2 is known to localize on the apical membrane in the epithelial cells of the renal proximal tubule and regulate reabsorption of glucose from the glomerular filtrate in the proximal tubule [25]. Meanwhile, SGLT2 localized on the mitochondria of Hep3B and Huh7 cells in this study. Recently, Villani et al. reported that CANA inhibits mitochondrial complex-I in prostate and lung cancer cells [28]. Taken together, these data indicate that SGLT2 may be involved in the mitochondrial function of Hep3B and Huh7 cells.

In our study, we mainly used 10 µM of CANA. In the pharmacokinetic study of CANA of healthy participants, absolute bioavailability concentration was reported to be approximately 3 to 15 µM [29]. Moreover, Kaji et al. used 10 µM of CANA as a clinically comparable dose in in vitro study using Huh7 and HepG2 cells [11]. Thus, 10 µM of CANA is thought to be clinical relevance of the concentration. In this study, 10 µM of CANA caused inhibition of cell proliferation, but not apoptosis, in Hep3B cells. Meanwhile, Kaji et al. previously reported that 10 µM of CANA causes apoptosis in HepG2 cells [11]. It remains unclear why our results were different from the previous results; however, a possibility is the difference in HCC cell lines between the studies. Wild type p53 occurs in HepG2 cells, while we used a Hep3B cell, which is a p53-null HCC cell line. In our study, apoptosis was evaluated by several assays including trypan blue staining, immunostaining for AN and 7AAD, and apoptosis-related molecules such as cleaved PARP. However, apoptosis was not detected in any of these assays in Hep3B and Huh7 cells. In addition, CANA caused G2/M arrest in this study. Thus, we showed that CANA caused inhibition of cell proliferation, but not apoptosis, in these hepatoma cell lines.

To investigate mechanisms for CANA-induced suppression of cell proliferation, we investigated changes in metabolite levels and protein expression of metabolic enzymes using metabolomics and iMPAQT. These two global analyses revealed that CANA did not alter the intracellular glucose level, although CANA is an SGLT2 inhibitor. The reason for this result remains unclear; however, we found that SGLT1 and various GLUT isoforms occurred in Hep3B and Huh7 cells. Since glucose metabolism plays crucial roles in cancer cell viability [30, 31], these glucose transporters may play a role in maintaining glucose homeostasis in Hep3B and Huh7 cells.

Meanwhile, we found that CANA downregulated proteins associated with the electron transport system in Hep3B cells. We also showed that CANA upregulated p-AMPKα1 and downregulated p-AMPKα2. The similar findings were also seen in Huh7 cells. CANA is

reported to inhibit electron transport systems and suppress proliferation of prostate cancer cells [28, 32]. In addition, CANA is reported to suppress ATP production in hepatoma cells [11]. Furthermore, CANA is reported to activate hepatic AMPK without alteration of insulin and glucagon signaling [33]. AMPK is known to be activated by up-regulation of p-AMPKα1 and downregulation of p-AMPKα2 [34, 35]. Thus, CANA may impair the mitochondrial electron transport system and ATP production, leading to AMPK activation via up-regulation of p-AMPKα1 and downregulation of p-AMPKα2 (Fig 7).

In this study, in Hep3B and Huh7 cells we found that CANA upregulated p-ACC, which is a downstream molecule of AMPK. Recently, ACC phosphorylation is reported to inhibit hepatic de novo lipogenesis and HCC proliferation [36]. We also found that CANA downregulated SCD, which is also a downstream molecule of AMPK. Down-regulation of SCD is known to suppress cell proliferation through regulation of monounsaturated fatty acids in prostate cancer cells [37]. Moreover, we found that CANA caused G2/M arrest of Hep3B cells. AMPK is also known to induce G2/M arrest via regulation of p53 and p21 in HepG2 cells [38, 39]. Taken together, CANA might inhibit oxidative phosphorylation and phosphorylation of AMPK, which results in suppression of cell proliferation through the following 3 pathways: 1) ACC phosphorylation, 2) down-regulation of SCD, and 3) G2/M arrest in Hep3B cells (Fig 7).

In our study, we further revealed that CANA affected fatty acid elongation including beta oxidation and up-regulation of butyrylcarnitine, acetylcarnitine, and 3-hydroxybutirate in Hep3B cells. Meanwhile, Liu et al. performed metabolomics and reported that synthesis of ketone bodies and fatty acid oxidation were upregulated in patients with HCC [40]. Ketone bodies are reported to suppress growth of colon and breast cancer cell lines through overexpression of uncoupling protein-2 [41]. Ketone supplementation is reported to decrease tumor cell viability and prolong survival of mice with metastatic brain tumor [42]. Moreover, in iMPAQT, we demonstrated that CANA down-regulated ACAA1, which is a key enzyme regulating beta-oxidation and production of ketone bodies. Since ACAA1 is reported to

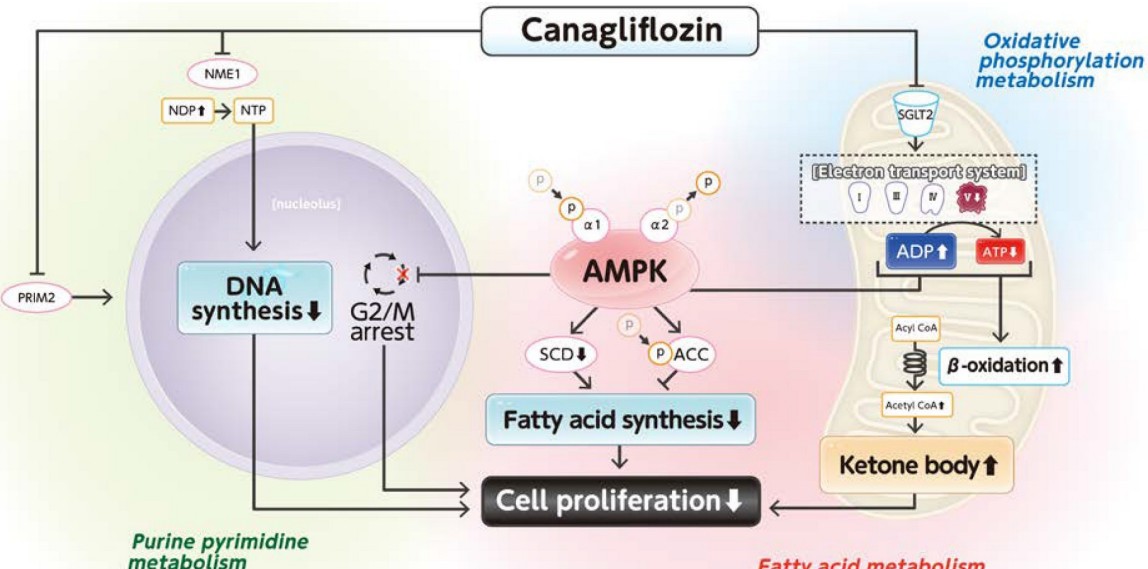

**Fig 7. A scheme for proposed mechanisms for canagliflozin-induced suppression of cell proliferation in Hep3B cell.** Abbreviations: SGLT2, sodium-glucose cotransporter 2; ADP, adenosine diphosphate; ATP, adenosine triphosphate; AMPK, AMP-activated protein kinase; SCD, stearoyl-CoA desaturase; ACC, acetyl-CoA carboxylase; NME1, nucleoside diphosphate kinase1; NDP, nucleotide diphosphate; NTP, nucleotide triphosphate; PRIM2, DNA primase, polypeptide 2; DNA, deoxyribonucleic acid.

promote the occurrence and progression of HCC [43, 44], suppression of ACAA1 may inhibit cell proliferation of HCC. Taken together, inhibition of cell proliferation of Hep3B may be caused by CANA-induced alterations in both metabolites and enzymes of fatty acid metabolism (Fig 7).

In this study, we first found that CANA suppressed enzymes regulating nucleotide synthesis using iMPAQT in Hep3B cells. CANA downregulated NME1, an enzyme catalyzing synthetic reaction of uridine triphosphate from UDP [45]. Moreover, in metabolomics, CANA upregulated UDP. NME1 and NME2 are reported to be upregulated in a mouse model of HCC [46]. Thus, these data indicate that CANA downregulates NME1 and inhibits RNA and DNA synthesis, leading to suppression of cell proliferation of Hep3B cells (Fig 7). Moreover, we found a down-regulation of PRIM2, (which encodes a subunit of primase involved in purine and pyrimidine metabolism), DNA replication, and transcription [47]. Thus, CANA may suppress cell proliferation of Hep3B cells by inhibiting of DNA synthesis by down-regulation of NME1 and PRIM2 (Fig 7). In metabolic analysis and iMPAQT analysis were performed only in Hep3B cells and the result may be specific in this cell line. The validation study using the various HCC cell lines is required.

It is unclear that the alterations in of hepatoma cells is CANA specific. In order to solve this issue, we examined the effect of dapagliflozin (10μM), a SGLT2 inhibiter, on cell proliferation in Hep3B. There was no significant difference in cell number between the dapagliflozin and control groups. In previous studies, Obara et al. reported that tofogliflozin, a SGLT2 inhibiter, did not suppress cell number in Huh7 and JHH cells [26]. Hung MH et al. reported that CANA specifically inhibited cell proliferation due to β-catenin-related pathway, which was not seen in the dapagliflozin and empagliflozin groups in Huh7 and Hep3B [48]. Thus, these previous reports along with our additional data suggest that suppression of cell proliferation in hepatoma cells may be CANA specific effect.

There are several limitations in this study. First limitation is that the impact of alterations in valine, leucine, and isoleucine metabolism remains unclear. There were significant alterations in 14 metabolites and 3 enzymes associated with valine, leucine, and isoleucine metabolism; however, these changes were dispersed and were not associated with a specific pathway in this study. Second, we did not show direct evidence the link between altered pathways and reduced cell proliferation. CANA mainly altered following pathways 1) oxidative phosphorylation/fatty acid metabolism pathway through down-regulation of ATP synthase F1 subunit alpha and 2) purine and pyrimidine metabolism though down-regulation of NME1 and PRIM2. In order to proof the direct evidence, up-regulation of ATP synthase F1 subunit alpha, NME1, and PRIM2 thought to be required. However, there is no available drugs, which activate these three molecules simultaneously. Third, it remains unclear if the effects of CANA on cell proliferation, metabolomics, and proteomics are cancer specific and are SGLT2 dependent. In this study, we could not examine the CANA specificity by knock down of SGLT2 protein by using small interfering RNA in Hep3B cells. However, Huang H et al. previously reported that a SGLT2i did not affect cell proliferation in normal human renal cells [49]. Kaji et al. reported that CANA did not suppress cell proliferation of HLE cells, which were SGLT2-negative cells [11]. However, we do not have normal hepatocytes, which proliferate in vitro and SGLT-2 negative cells. Further study will be focused on the impact of amino acids metabolism on cell proliferation, direct evidence between altered pathways and CANA, and cell specificity of CANA.

Further study is required to investigate the impact of altered valine, leucine, and isoleucine metabolism on CANA-caused suppression of HCC proliferation.

In conclusion, SGLT2 was expressed and localized on mitochondria in Hep3B and Huh7 cells. CANA significantly suppressed proliferation of these cells. Multi-omics analysis of metabolomics and iMPAQT revealed that CANA mainly altered 1) oxidative phosphorylation

metabolism, 2) fatty acid metabolism, and 3) purine and pyrimidine metabolism, but not glucose metabolism. CANA also altered phosphorylation of AMPK and ACC, which are sensors of intracellular ATP levels and regulators for beta oxidation. Thus, CANA may suppress proliferation of hepatoma cells via regulation of the electron transport system, beta oxidation, and nucleic acid synthesis.

## Supporting information

**S1 Fig. Immunoblotting for SGLT2 in Hep3B and Huh7.** Abbreviations: CANA, canagliflozin; SGLT1, sodium-glucose cotransporter 1; GLUT, glucose transporter.
(TIFF)

**S2 Fig. Immunoblotting for SGLT1 and GLUT1-6 in Hep3B and Huh7.** Abbreviations: CANA, canagliflozin; SGLT1, sodium-glucose cotransporter 1; GLUT, glucose transporter.
(TIFF)

**S3 Fig. Effects of dapagliflozin on proliferation of Hep3B cells.** * P<0.01. Abbreviations: CON, control; DAPA, dapagliflozin.
(TIFF)

**S4 Fig. Effects of CANA on morphological change in Hep3B and Huh7 cells.** Scale bar = 50 μm. Abbreviations: CON, control; CANA, canagliflozin.
(TIFF)

**S5 Fig. Immunoblotting for apoptosis-related molecules in Hep3B and Huh7 cells.** Abbreviations: CON, control; CANA, canagliflozin; PARP, poly adenosine diphosphate-ribose polymerase.
(TIFF)

**S6 Fig. Immunoblotting for fatty acid metabolism-associated molecules in Huh7 cells.** Abbreviations: CON, control; CANA, canagliflozin; AMPK, AMP-activated protein kinase; ACC, acetyl-CoA carboxylase.
(TIFF)

**S7 Fig. Metabolism map for valine, leucine, and isoleucine metabolism.** Red line indicates an up-regulated pathway. Red circle indicates an up-regulated metabolite. Blue circle indicates a down-regulated metabolite.
(TIFF)

**S8 Fig. Intensity of protein expression in the 10 μM CANA and CON groups.** Abbreviations: CON, control; CANA, canagliflozin; AMPK, AMP-activated protein kinase; ACC, acetyl-CoA carboxylase.
(TIFF)

**S1 Raw image.**
(PDF)

**S1 Table. Effects of CANA on levels of 225 metabolites by metabolomics in Hep3B cells.**
(DOCX)

**S2 Table. Effects of CANA on expression level of 342 metabolic enzymes by iMPAQT assay in Hep3B cells.**
(DOCX)

## Acknowledgments

We thank Ms. Saori Meifu Division of Gastroenterology, Department of Medicine, Kurume University School of Medicine) for technical assistance with flow cytometry. We would like to thank Editage for English language editing.

## Author Contributions

**Conceptualization:** Dan Nakano, Takumi Kawaguchi, Hideki Iwamoto, Masako Hayakawa, Hironori Koga, Takuji Torimura.

**Data curation:** Dan Nakano, Takumi Kawaguchi, Hideki Iwamoto, Masako Hayakawa, Hironori Koga, Takuji Torimura.

**Funding acquisition:** Takumi Kawaguchi.

**Investigation:** Hideki Iwamoto, Masako Hayakawa.

**Methodology:** Dan Nakano, Hironori Koga, Takuji Torimura.

**Project administration:** Takumi Kawaguchi.

**Supervision:** Takumi Kawaguchi, Hideki Iwamoto, Hironori Koga, Takuji Torimura.

**Validation:** Takumi Kawaguchi, Masako Hayakawa.

**Visualization:** Takumi Kawaguchi, Masako Hayakawa, Takuji Torimura.

**Writing – original draft:** Dan Nakano, Takumi Kawaguchi.

**Writing – review & editing:** Hironori Koga, Takuji Torimura.

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
