## [Decision Letter · Decision Letter 0]

24 Dec 2019

PONE-D-19-33203

Effects of Canagliflozin on Growth and Metabolic Reprograming in Hepatocellular Carcinoma Cells: Multi-Omics Analysis of Metabolomics and Absolute Quantification Proteomics (iMPAQT)

PLOS ONE

Dear Dr. Takumi Kawaguchi,

Thank you for submitting your manuscript to PLOS ONE. After careful consideration, we feel that it has merit but does not fully meet PLOS ONE’s publication criteria as it currently stands. Therefore, we invite you to submit a revised version of the manuscript that addresses the points raised during the review process.

We would appreciate receiving your revised manuscript by Feb 07 2020 11:59PM. To enhance the reproducibility of your results, we recommend that if applicable you deposit your laboratory protocols in protocols.io, where a protocol can be assigned its own identifier (DOI) such that it can be cited independently in the future. For instructions see: http://journals.plos.org/plosone/s/submission-guidelines#loc-laboratory-protocols

We look forward to receiving your revised manuscript.

Kind regards,

Tatsuo Kanda, M.D., Ph.D.

Academic Editor

PLOS ONE

Journal Requirements:

2. Please provide additional information about the Huh7 and Hep3B cell lines used in this work, including  the source, history any quality control testing procedures (authentication, characterisation, and mycoplasma testing). For more information, please see http://journals.plos.org/plosone/s/submission-guidelines#loc-cell-lines.

3. To comply with PLOS ONE submission guidelines, in your Methods section, please provide additional information regarding your statistical analyses. For more information on PLOS ONE's expectations for statistical reporting, please see https://journals.plos.org/plosone/s/submission-guidelines.#loc-statistical-reporting.

4. We suggest you thoroughly copyedit your manuscript for language usage, spelling, and grammar. If you do not know anyone who can help you do this, you may wish to consider employing a professional scientific editing service.  

5. At this time, we ask that you please include scale bars on the microscopy images presented in Figures 1 and 3 and refer to the scale bar in the corresponding Figure legend.

6. Please expand the acronym “AMED” (as indicated in your financial disclosure) so that it states the name of your funders in full.

7. Thank you for stating the following in the Competing Interests section:

'Takumi Kawaguchi received lecture fees from Mitsubishi Tanabe Pharma Corporation, MSD K.K., and Otsuka Pharmaceutical Co., Ltd. The other authors have no conflicts of interest'

8. PLOS ONE now requires that authors provide the original uncropped and unadjusted images underlying all blot or gel results reported in a submission’s figures or Supporting Information files. This policy and the journal’s other requirements for blot/gel reporting and figure preparation are described in detail at https://journals.plos.org/plosone/s/figures#loc-blot-and-gel-reporting-requirements and https://journals.plos.org/plosone/s/figures#loc-preparing-figures-from-image-files. When you submit your revised manuscript, please ensure that your figures adhere fully to these guidelines and provide the original underlying images for all blot or gel data reported in your submission. See the following link for instructions on providing the original image data: https://journals.plos.org/plosone/s/figures#loc-original-images-for-blots-and-gels.

Reviewers' comments:

Reviewer's Responses to Questions

**Comments to the Author**

1. Is the manuscript technically sound, and do the data support the conclusions?

Reviewer #1: Yes

Reviewer #2: Partly

Reviewer #3: No

2. Has the statistical analysis been performed appropriately and rigorously? 

Reviewer #1: Yes

Reviewer #2: I Don't Know

Reviewer #3: Yes

3. Have the authors made all data underlying the findings in their manuscript fully available?

Reviewer #1: Yes

Reviewer #2: No

Reviewer #3: Yes

4. Is the manuscript presented in an intelligible fashion and written in standard English?

Reviewer #1: Yes

Reviewer #2: No

Reviewer #3: Yes

5. Review Comments to the Author

Reviewer #1: The authors tried to evaluate the effects of Canagliflozin (CANA) on growth and metabolic reprogramming in HCC cells and found that the CANA suppressed the proliferation of HCC cells through alterations in mitochondrial oxidative phosphorylation metabolism, fatty acid metabolism, and purine and pyrimidine metabolism. The paper is well written and organized. They designed the study based on the previous findings of SGLT2 inhibitor’s anti-tumor effect and further evaluated the direct mechanism toward HCC cells using iMPAQT technique.

Comments;

CANA (10uM) was used for metabolomics experiment. Was the significant decrease of 30uM due to cytotoxic effect of the drug? Have the authors studied any of the significant proteins in CANA (30uM), i.e., was there any dose dependency in protein expression?

SGLT2 was expressed in the mitochondria of Hep3B and Huh7 cells. Was SGLT2 expressed in normal hepatocyte or only in cancer cells? Expression levels in normal hepatocyte and its localization will be of great interest. If possible, CANA’s effect on commercially available human primary hepatocyte culture or PXB mouse primary culture could be studied.

Some of the experiments were only studied in Hep3B cell. For instance, how was the cell cycle experiment in Huh7 cell?

Was there any morphological change after CANA treatment?

Reviewer #2: In the present study, the anti-cancer role of SGLT2 inhibitor, canagliflozin (CANA) is examined using SGLT2 expressing liver cancer cell lines. Several metabolic pathways were affected by canagliflozin using combined analysis of metabolites and metabolic enzymes. Mitochondrial localization of SGLT2 is suggested to have some role. Although the findings are potentially interesting, there are several points which need to be addressed.

1. Major point is that although CANA affected metabolome and several specific pathways, no direct evidence is presented for the link between altered pathways and reduced cell proliferation.

2. In Fig. 1a, did the authors confirm if the protein band is SGLT2 specific by using SGLT2-positive and negative control samples?

3. In Fig. 1c, although the immunofluorescence experiments indicate colocalization of SGLT2 and mitochondria, the results are not very convincing. SGLT2 distribution could be confirmed by immunoblot after isolating mitochondrial fraction from cytosolic fraction.

4. In Fig. 2, if the reduced cell proliferation by CANA treatment is dependent on SGLT2 expression, the proliferation of cells not expressing SGLT2 is less affected by CANA treatment.

5. Please mention the clinical relevance of the concentration of the drug used in the present study. Does the drug also affect the normal cell proliferation?

6. Please mention whether the effect of CANA on the mitochondrial function and metabolome is cancer specific and is SGLT2 dependent.

Did the authors examine whether the identified molecules from metabolome analysis were altered in both cell lines used, and other cells such as SGLT2-negative cells and kidney cells?

7. Please specify the cell line used in the experiment in each figure legend for clarity. Please mention the cell line and treatment protocol used in the metabolomics and “iMPACT” analysis in the method section.

8. Please provide all the measurement results in the metabolome and proteome analysis.

Reviewer #3: In this article, the author assessed the effects of canagliflozin (CANA) on proliferation and metabolic reprograming of HCC cell line. They revealed that CANA reduced proliferation of Hep3B and Huh7. They also showed CANA altered mitochondrial oxidative phosphorylation metabolism, fatty acid metabolism, and purine and pyrimidine metabolism in Hep3B

1) Figure 2C, D; Live cell rate is decreased in CANA 10uM group. However, dead cell numbers in CANA 10uM group are not different from those in CON group. It seems that there is discrepancy between these data. Is dead cell rate in CANA 10uM group same with CON group?

2) In the study using iMPAQT, the dose of CANA should be described.

3) Metabolic analysis and iMPAQT analysis is performed only in Hep3B cells. The result may be specific in this cell line. The validation using the other cell line should be included.

4) The data is not shown the alteration in mitochondrial oxidative phosphorylation metabolism, fatty acid metabolism, and purine and pyrimidine metabolism really affects cell proliferation.

6. PLOS authors have the option to publish the peer review history of their article (what does this mean?). If published, this will include your full peer review and any attached files.

Reviewer #1: No

Reviewer #2: No

Reviewer #3: No

---

## [Author Response · Author response to Decision Letter 0]

11 Feb 2020

Responses to the Academic Editor

Thank you for your comments regarding our manuscript (Manuscript PONE-D-19-33203). We appreciate your comments, which have helped us to improve our manuscript. In line with the comments, please find below our point-by-point responses.

Answer: We ensure that our manuscript meets PLOS ONE's style requirements, including those for file naming.

2) Please provide additional information about the Huh7 and Hep3B cell lines used in this work, including the source, history any quality control testing procedures (authentication, characterisation, and mycoplasma testing).

Answer: We apologize that we did not provide sufficient information about cell lines. Huh7 cells were obtained from HuH7 (JCRB0403) and HLF (JCRB0405) cells were obtained from the JCRB Cell Bank (Tokyo, Japan). Hep3B cells (HB8064) were obtained from the American Type Culture Collection (ATCC-LGC Standards). HAK-1A, HAK-1B, KYN-2, and KMCH-1 were kindly provided from Prof. Hirohisa Yano (Department of Pathology, Kurume University School of Medicine, Kurume, Japan). These cells were checked by quality control testing procedures including authentication, characterisation, and mycoplasma testing. The above information was added in the revised manuscript (Page 7, line 23-Page 8, line 5).

3) To comply with PLOS ONE submission guidelines, in your Methods section, please provide additional information regarding your statistical analyses.

Answer: We apologize that we did not provide sufficient information regarding our statistical analyses. We added additional information regarding our statistical analyses in the revised manuscript (Page 15, line 17-19).

4) We suggest you thoroughly copyedit your manuscript for language usage, spelling, and grammar. If you do not know anyone who can help you do this, you may wish to consider employing a professional scientific editing service.

Answer: This manuscript was edited by Editage. The certificate of English editing was attached in this response letter. In the revised manuscript, following sentence was added “We would like to thank Editage for English language editing.” (Page 39, line 5).

5) At this time, we ask that you please include scale bars on the microscopy images presented in Figures 1 and 3 and refer to the scale bar in the corresponding Figure legend.

Answer: We apologize that we did not provide scale bar. In the revised manuscript, the scale bars and the corresponding figure legends were added (Figures 1B, 1C, 3D, 3E, 3F, and Supplementary Figure 3).

6) Please expand the acronym “AMED” (as indicated in your financial disclosure) so that it states the name of your funders in full.

Answer: We apologize that we did not spell out the acronym “AMED”. In the revised manuscript, the acronym “AMED” was expanded as following: “This research was supported by Japan Agency for Medical Research and Development (AMED) under Grant Number JP19fk0210040.” (Page 39, line 17-18).

7) Thank you for stating the following in the Competing Interests section: 'Takumi Kawaguchi received lecture fees from Mitsubishi Tanabe Pharma Corporation, MSD K.K., and Otsuka Pharmaceutical Co., Ltd. The other authors have no conflicts of interest' Please confirm that this does not alter your adherence to all PLOS ONE policies on sharing data and materials, by including the following statement: "This does not alter our adherence to PLOS ONE policies on sharing data and materials.”

Answer: Takumi Kawaguchi received lecture fees from Mitsubishi Tanabe Pharma Corporation, MSD K.K., and Otsuka Pharmaceutical Co., Ltd. However, this does not alter our adherence to PLOS ONE policies on sharing data and materials. The description was added in the revised main text as well as the cover letter. (Page 39, line 8-11).

8) Please know it is PLOS ONE policy for corresponding authors to declare, on behalf of all authors, all potential competing interests for the purposes of transparency. PLOS defines a competing interest as anything that interferes with, or could reasonably be perceived as interfering with, the full and objective presentation, peer review, editorial decision-making, or publication of research or non-research articles submitted to one of the journals. Competing interests can be financial or non-financial, professional, or personal. Competing interests can arise in relationship to an organization or another person.

Answer: We appreciate for letting us know the PLOS ONE policy. We understand the policy.

9) PLOS ONE now requires that authors provide the original uncropped and unadjusted images underlying all blot or gel results reported in a submission’s figures or Supporting Information files.

Answer: We appreciate for letting us know the requirement for the original uncropped and unadjusted images underlying all blot or gel results. We understand the requirements.

----------

Responses to REVIEWER 1,

Thank you for your comments regarding our manuscript (Manuscript PONE-D-19-33203). We appreciate your comments, which have helped us to improve our manuscript. In line with the comments, please find below our point-by-point responses.

1) CANA (10uM) was used for metabolomics experiment. Was the significant decrease of 30uM due to cytotoxic effect of the drug? Have the authors studied any of the significant proteins in CANA (30uM), i.e., was there any dose dependency in protein expression?

Answer: Following your suggestion, we investigated effects of 30 μM of CANA on cell number, morphological changes, and apoptosis in Hep3B and Huh7 cells. At 72 hours after treatment, the number of cells was significantly decreased in the 30 μM CANA group compared to the CON, 3 μM, and 10 μM CANA groups with a dose-dependent fashion (Figure 2A). In phase-contrast-microscopic image, suspended cells were seen in the 30 μM CANA group, while suspended cells were not seen in the CON, 3 μM, and 10 μM CANA groups in Hep3B and Huh7 cells 48 hours after treatment (S3 Fig). In trypan-blue staining assay, the number of dead cells were significantly increased in the 30 μM CANA group compared to that in the CON, and 10 μM CANA groups in Hep3B cells 48 hours after treatment (Figure 2D). Furthermore, upregulation of cleaved Poly (adenosine diphosphate-ribose) polymerase was only seen in 30 μM of CANA, but not in the CON and 10μM CANA groups, indicating the dose dependency of CANA in apoptosis-related protein expression in Hep3B and Huh7 cells (S4A and B Figs). Thus, these results suggested that treatment with 30 μM of CANA significantly decreased cell number due to cytotoxic effect of the drug. The above descriptions were added to the revised manuscript (Page 18, line 8–13; Page 18, line 23-Page 19, line 9).

2) SGLT2 was expressed in the mitochondria of Hep3B and Huh7 cells. Was SGLT2 expressed in normal hepatocyte or only in cancer cells? Expression levels in normal hepatocyte and its localization will be of great interest. If possible, CANA’s effect on commercially available human primary hepatocyte culture or PXB mouse primary culture could be studied.

Answer: We appreciate your comments. As you suggested, we investigated expression of SGLT2 in human primary hepatocytes (LHum17003; BIOPREDIC, Saint-Grégoire, France) and found that expression of SGLT2 was weak (Fig 1D). We have added the results in the revised manuscript (Page 16, line 4-10).

 It is great interest to investigate effects of CANA on human primary hepatocyte culture or PXB mouse primary culture. However, we do not have normal hepatocytes, which proliferate in vitro and could not investigate effects of CANA on the proliferation of normal cells. Therefore, this issue was described as a limitation of this study (Page 38, line 10-11).

3) Some of the experiments were only studied in Hep3B cell. For instance, how was the cell cycle experiment in Huh7 cell

Answer: Following your suggestion, we investigated effects of CANA on expression of apoptosis-related molecules and cell cycle in Huh7 cells. Like in Hep3B cells, there was no significant difference in the expression of apoptosis-related molecules including cleaved PARP between the 10 μM CANA and CON groups in Huh7 cells. (S2 Fig A and B). Furthermore, in Huh7 cells, the percentage of G2/M phase was also significantly increased to 12.5±0.2% and 14.0±0.2% in the 10 μM and 30 μM CANA groups (both P<0.01), respectively (Fig 3G). Thus, these experiments in Huh7 cells yielded similar results in Hep3B. These results were added in the revised manuscript (Page 19, line 5–9; Page 20, line 1–4).

4) Was there any morphological change after CANA treatment? 

Answer: We appreciate for your valuable comment. Following your suggestion, we examined morphological change of Hep3B and Huh7 cells after CANA treatment by using phase-contrast-microscope. There was no morphological change in Hep3B and Huh 7 cells after 10 μM of CANA treatment (Supplementary Figure 3). However, after treatment of 30 μM CANA, there was morphological changes such as spindled and/or rounded shapes in Hep3B and Huh 7 cells (Supplementary Figure 3). These results were added in the revised manuscript (Page 18, line 8-13).

----------

Responses to REVIEWER 2,

Thank you for your comments regarding our manuscript (Manuscript PONE-D-19-33203). We appreciate your comments, which have helped us to improve our manuscript. In line with the comments, please find below our point-by-point responses.

1) Major point is that although CANA affected metabolome and several specific pathways, no direct evidence is presented for the link between altered pathways and reduced cell proliferation.

Answer: We totally agree with your comment. We did not present direct evidence for the link between altered pathways and reduced cell proliferation in this study. In this study, CANA mainly altered following pathways 1) oxidative phosphorylation/fatty acid metabolism pathway through down-regulation of ATP synthase F1 subunit alpha and 2) purine and pyrimidine metabolism though down-regulation of NME1 and PRIM2. In order to proof the direct evidence, up-regulation of ATP synthase F1 subunit alpha, NME1, and PRIM2 thought to be required. However, there is no available drugs, which activate these three molecules simultaneously. Thus, it seems impossible to proof the direct evidence currently. As far as we searched, there is no study, which proof the direct evidence for the link between altered pathways and reduced cell proliferation by up-/down-regulation of multiple pathways [1]. Therefore, we described this issue as a limitation of this study (Page 37, line 22-Page 38, line 5).

2) In Fig. 1a, did the authors confirm if the protein band is SGLT2 specific by using SGLT2-positive and negative control samples?

Answer: We appreciate for your comment. We selected Jurkat cells as a SGLT2-positive sample as previously described [2] and the protein band was detected at the same molecule size of the protein band in Hep3B and Huh7 cells. Although we examined the expression of SGLT2 in 8 hepatoma cell lines and human primary hepatocytes (LHum17003; BIOPREDIC, Saint-Grégoire, France). However, expression of SGLT2 was seen in the all hepatoma cell lines. Even in human primary hepatocytes, weak expression of SGLT2 was observed. Thus, we could not find out SGLT2-negative sample in this study. These results were added in the revised manuscript (Page 16, line 7-10).

3) In Fig. 1c, although the immunofluorescence experiments indicate colocalization of SGLT2 and mitochondria, the results are not very convincing. SGLT2 distribution could be confirmed by immunoblot after isolating mitochondrial fraction from cytosolic fraction.

Answer: Following your suggestion, SGLT2 distribution was investigated by immunoblot after isolating mitochondrial fraction from cytosolic fraction. In Hep3B and Huh7 cells, marked expression of SGLT2 was detected in mitochondrial fraction. On the other hand, expression of SGLT2 was weak cytoplasmic fraction. These data suggested that SGLT2 localized in mitochondria in Hep3B and Huh7 cells (Fig 1D). These results were added in the revised manuscript (Page 16, line 14-16).

4) In Fig. 2, if the reduced cell proliferation by CANA treatment is dependent on SGLT2 expression, the proliferation of cells not expressing SGLT2 is less affected by CANA treatment.

Answer: We appreciate for your valuable comment. As you suggested, we investigated expression of SGLT2 by western blotting and expression of SGLT2 was seen in 8 hepatoma cell lines such as Huh7, HLF, HepG2, Hep3B, KYN2, KMCH1, HAK1A, and, HAK1B cell lines (Supplementary Figure 1). On the other hand, Kaji et al. previously reported that expression of SGLT2 was not seen in HLE cells, a hepatoma cell line [2]. Moreover, they showed that CANA did not suppress cell proliferation of HLE at 10 μM CANA, which is same concentration of our study. These data suggested that effects of CANA treatment on cell proliferation may be dependent on SGLT2 expression. However, we do not have normal hepatocytes, which proliferate in vitro, and, therefore, this issue was described as a limitation of this study (Page 16, line 4–7; Page 38, line 5-11).

5) Please mention the clinical relevance of the concentration of the drug used in the present study. Does the drug also affect the normal cell proliferation?

Answer: We agree with your comment. We did not mention the clinical relevance of the concentration of the drug used in the study. In the pharmacokinetic study of CANA, absolute bioavailability concentration was reported to be approximately 3 to 15 μM [3]. Moreover, Kaji et al. used 10 μM of CANA as a clinically comparable dose in in vitro study using Huh7 and HepG2 cells [2]. We also used 10 μM of CANA in this study, which is thought to be clinical relevance of the concentration. The above description has been added to the revised manuscript (Page 33, line 23-Page 34, line 5).

 As you indicated, it is important to investigate effects of CANA on cell proliferation in normal cell. Huang H et al. previously reported that dapagliflozin, a SGLT2i, did not affect cell proliferation in normal human renal cells [4]. However, we do not have normal hepatocytes, which proliferate in vitro, and, therefore, this issue was described as a limitation of this study (Page 38, line 5-11).

6) Please mention whether the effect of CANA on the mitochondrial function and metabolome is cancer specific and is SGLT2 dependent.

Did the authors examine whether the identified molecules from metabolome analysis were altered in both cell lines used, and other cells such as SGLT2-negative cells and kidney cells?

Answer: We appreciate for your valuable comments. Huang H et al. previously reported that dapagliflozin, a SGLT2i, exerts cytotoxic effect in human RCC cell lines, but not in normal human renal cells [4]. However, in this study, we did not evaluate effect of CANA on the mitochondrial function and metabolome in normal hepatocytes and therefore it remains unclear if effects of CANA on the mitochondrial function and metabolome is cancer specific. Moreover, Kaji et al. previously reported that HLE cells, a hepatoma cell line, were SGLT2-negative and showed that cell proliferation of HLE cells was not suppressed by treatment with 10 μM of CANA [2], which is same concentration of our study. These data suggested that effects of CANA treatment on cell proliferation may be dependent on SGLT2 expression. However, we do not have normal hepatocytes, which proliferate in vitro and SGLT2-negative cells. Therefore, this issue was described as a limitation of this study (Page 38, line 5-11).

7) Please specify the cell line used in the experiment in each figure legend for clarity. Please mention the cell line and treatment protocol used in the metabolomics and “iMPACT” analysis in the method section.

Answer: We apologize for insufficient description of this issue. Following your suggestion, we specified the cell line used in the experiment in the main text and each figure legend throughout the manuscript. We also specified the cell line and treatment protocol used in the metabolomics and “iMPACT” analysis in the method section (Page 13, line 18, Page 14, line 12-13).

8) Please provide all the measurement results in the metabolome and proteome analysis.

Answer: We apologize for insufficient description of the results. Following your suggestion, we provided all the measurement results in the metabolomics and “iMPACT” analysis. In the main text, we presented the results for significantly altered metabolites and metabolic enzymes because all the data occupies a large amount of space. All the measurement results were presented as Supplementary Tables (S1 and 2 Tables). 

References

1. Yu L, Wu J, Zhai Q, Tian F, Zhao J, Zhang H, et al. Metabolomic analysis reveals the mechanism of aluminum cytotoxicity in HT-29 cells. PeerJ. 2019;7:e7524. doi: 10.7717/peerj.7524. PubMed PMID: 31523502; PubMed Central PMCID: PMCPMC6716502.

2. Kaji K, Nishimura N, Seki K, Sato S, Saikawa S, Nakanishi K, et al. Sodium glucose cotransporter 2 inhibitor canagliflozin attenuates liver cancer cell growth and angiogenic activity by inhibiting glucose uptake. International journal of cancer. 2018;142(8):1712-22. doi: 10.1002/ijc.31193. PubMed PMID: 29205334.

3. Devineni D, Murphy J, Wang SS, Stieltjes H, Rothenberg P, Scheers E, et al. Absolute oral bioavailability and pharmacokinetics of canagliflozin: A microdose study in healthy participants. Clin Pharmacol Drug Dev. 2015;4(4):295-304. doi: 10.1002/cpdd.162. PubMed PMID: 27136910.

4. Kuang H, Liao L, Chen H, Kang Q, Shu X, Wang Y. Therapeutic Effect of Sodium Glucose Co-Transporter 2 Inhibitor Dapagliflozin on Renal Cell Carcinoma. Med Sci Monit. 2017;23:3737-45. doi: 10.12659/msm.902530. PubMed PMID: 28763435; PubMed Central PMCID: PMCPMC5549715.

----------

Responses to REVIEWER 3,

Thank you for your comments regarding our manuscript (Manuscript PONE-D-19-33203). We appreciate your comments, which have helped us to improve our manuscript. In line with the comments, please find below our point-by-point responses.

1) Figure 2C, D; Live cell rate is decreased in CANA 10uM group. However, dead cell numbers in CANA 10uM group are not different from those in CON group. It seems that there is discrepancy between these data. Is dead cell rate in CANA 10uM group same with CON group?

Answer: We apologize for wrong description in vertical line in Figure 2C. In the figure, we examined number of live cells in CANA 10 μM by comparing to the CON group and the result was expressed by %CON. We corrected this mistake in the revised manuscript (Figure 2C and D). Again, we apologize for causing confusion with the wrong description.

2) In the study using iMPAQT, the dose of CANA should be described.

Answer: We apologize for insufficient description in the study using iMPAQT. The dose of CANA (10 μM) was added in the revised manuscript (Page 14, line 12-13).

3) Metabolic analysis and iMPAQT analysis is performed only in Hep3B cells. The result may be specific in this cell line. The validation using the other cell line should be included.

Answer: We totally agree with your comment. Expression of SGLT2 was seen in both Hep3B and Huh 7 cells, and CANA inhibited cell proliferation of both HCC cell lines. We examined changes in phosphorylation of AMPKα1 and ACC and found that CANA phosphorylated AMPKα1 and ACC not only in Hep3B cell, but also in Huh 7 cells (S5 Fig). However, as you indicated, iMPAQT analysis was performed only in Hep3B cells. Therefore, the result may be specific in this cell line. The validation study using the various HCC cell lines should be performed. Unfortunately, we could not conduct the validation study at this moment. Therefore, this issue was described in the Discussion section of the revised manuscript (Page 32 line 7–9; Page 37 line 4-7).

4) The data is not shown the alteration in mitochondrial oxidative phosphorylation metabolism, fatty acid metabolism, and purine and pyrimidine metabolism really affects cell proliferation.

Answer: We totally agree with your comment. We did not present direct evidence for the link between altered pathways and reduced cell proliferation in this study. In this study, CANA mainly altered following pathways 1) oxidative phosphorylation/fatty acid metabolism pathway through down-regulation of ATP synthase F1 subunit alpha and 2) purine and pyrimidine metabolism though down-regulation of NME1 and PRIM2. In order to proof the direct evidence, up-regulation of ATP synthase F1 subunit alpha, NME1, and PRIM2 thought to be required. However, there is no available drugs, which activate these three molecules simultaneously. Thus, it seems impossible to proof the direct evidence currently. As far as we searched, there is no study, which proof the direct evidence for the link between altered pathways and reduced cell proliferation by up-/down-regulation of multiple pathways [1]. Therefore, we described this issue as a limitation of this study (Page 37, line 22-Page 38, line 5).

Reference

1. Yu L, Wu J, Zhai Q, Tian F, Zhao J, Zhang H, et al. Metabolomic analysis reveals the mechanism of aluminum cytotoxicity in HT-29 cells. PeerJ. 2019;7:e7524. doi: 10.7717/peerj.7524. PubMed PMID: 31523502; PubMed Central PMCID: PMCPMC6716502.

---

## [Decision Letter · Decision Letter 1]

25 Feb 2020

PONE-D-19-33203R1

Effects of Canagliflozin on Growth and Metabolic Reprograming in Hepatocellular Carcinoma Cells: Multi-Omics Analysis of Metabolomics and Absolute Quantification Proteomics (iMPAQT)

PLOS ONE

Dear Prof. Kawaguchi,

Thank you for submitting your manuscript to PLOS ONE. After careful consideration, we feel that it has merit but does not fully meet PLOS ONE’s publication criteria as it currently stands. Therefore, we invite you to submit a revised version of the manuscript that addresses the points raised during the review process.

We would appreciate receiving your revised manuscript by Apr 10 2020 11:59PM. To enhance the reproducibility of your results, we recommend that if applicable you deposit your laboratory protocols in protocols.io, where a protocol can be assigned its own identifier (DOI) such that it can be cited independently in the future. For instructions see: http://journals.plos.org/plosone/s/submission-guidelines#loc-laboratory-protocols

We look forward to receiving your revised manuscript.

Kind regards,

Tatsuo Kanda, M.D., Ph.D.

Academic Editor

PLOS ONE

Reviewers' comments:

Reviewer's Responses to Questions

**Comments to the Author**

1. If the authors have adequately addressed your comments raised in a previous round of review and you feel that this manuscript is now acceptable for publication, you may indicate that here to bypass the “Comments to the Author” section, enter your conflict of interest statement in the “Confidential to Editor” section, and submit your "Accept" recommendation.

Reviewer #1: All comments have been addressed

Reviewer #2: (No Response)

Reviewer #3: (No Response)

2. Is the manuscript technically sound, and do the data support the conclusions?

Reviewer #1: Yes

Reviewer #2: (No Response)

Reviewer #3: (No Response)

3. Has the statistical analysis been performed appropriately and rigorously? 

Reviewer #1: Yes

Reviewer #2: (No Response)

Reviewer #3: (No Response)

4. Have the authors made all data underlying the findings in their manuscript fully available?

Reviewer #1: Yes

Reviewer #2: (No Response)

Reviewer #3: (No Response)

5. Is the manuscript presented in an intelligible fashion and written in standard English?

Reviewer #1: Yes

Reviewer #2: (No Response)

Reviewer #3: (No Response)

6. Review Comments to the Author

Reviewer #1: The authors have fulfilled each of the major compulsory revisions and modified the manuscript as requested. I have the following further suggestion that in my opinion will improve the quality of the manuscript.

Figure 6 needs statistics (actual p-values).

Page 18 Line 6, Effect of CANA on apoptosis morphological change of Hep3B and Huh7 cells. Needs hyphenation or modification.

Reviewer #2: In the discussion, “it remains unclear if the effects of CANA on cell proliferation, metabolomics, and proteomics are cancer specific and are SGLT2 dependent.”

1. Authors could examine the effects of knockdown/knockout of SGLT2.

2. Please mention if the effects are CANA specific or common features of SGLT2 inhibitors.

Reviewer #3: (No Response)

7. PLOS authors have the option to publish the peer review history of their article (what does this mean?). If published, this will include your full peer review and any attached files.

Reviewer #1: No

Reviewer #2: No

Reviewer #3: No

---

## [Author Response · Author response to Decision Letter 1]

9 Apr 2020

Dear Prof. Tatsuo Kanda, M.D., Ph.D.

Thank you very much for your letter dated on February 26, 2016 regarding our manuscript (Manuscript # PONE-D-19-33203R1). We appreciate your comments and those of the reviewers, which have helped us to improve our manuscript.

Enclosed please find the revised manuscript. Our responses to the reviewer’s comments are described in the attached sheet and all changes are indicated with Track Changes in the revised manuscript. We hope that these revisions respond to your comments and the manuscript is now suitable to PLOS ONE.

This research was supported by Japan Agency for Medical Research and Development (AMED) under Grant Number JP19fk0210040. Takumi Kawaguchi received lecture fees from Mitsubishi Tanabe Pharma Corporation, MSD K.K., and Otsuka Pharmaceutical Co., Ltd. This does not alter our adherence to PLOS ONE policies on sharing data and materials.

Thank you very much for your kind consideration of our manuscript.

Sincerely Yours,

Takumi Kawaguchi, M.D., Ph.D.

Responses to REVIEWER 1,

Thank you for your comments regarding our manuscript (Manuscript PONE-D-19-33203). We appreciate your comments, which have helped us to improve our manuscript. In line with the comments, please find below our point-by-point responses.

1) Figure 6 needs statistics (actual p-values).

Answer: Following your suggestion, we measured intensity of protein expression in the 10 μM CANA and CON groups by using image J. The intensity of p-AMPKα1 in the 10 μM CANA groups was significantly higher than that in the CON group. On the other hand, the intensity of p-AMPKα2 in the 10 μM CANA groups was significantly lower than that in the CON group. These results were added in the Supplementary Figure 8. We appreciate for your suggestion, which have helped us to improve our manuscript. These results were added in the revised manuscript (Page 10, line 21-22).

2) Page 18 Line 6, Effect of CANA on apoptosis morphological change of Hep3B and Huh7 cells. Needs hyphenation or modification.

Answer: We apologized for the typo. As you suggested, we corrected for the grammatical error as following; Effect of CANA on morphological change and apoptosis in Hep3B and Huh7 cells (Page 18, line 9-10). 

Responses to REVIEWER 2,

Thank you for your comments regarding our manuscript (Manuscript PONE-D-19-33203). We appreciate your comments, which have helped us to improve our manuscript. In line with the comments, please find below our point-by-point responses.

Reviewer #2: 

1. In the discussion, “it remains unclear if the effects of CANA on cell proliferation, metabolomics, and proteomics are cancer specific and are SGLT2 dependent.” Authors could examine the effects of knockdown/knockout of SGLT2.

Answer: Following your suggestion, we performed knock down of SGLT2 using small interfering RNA (siRNA) (Silencer® Select s534032 Thermo Fisher), because knock down is reported to be suitable for investigating phenotypic change than knock out [1, 2].

 We transfected siRNA for SGLT2 into Hep3B cells according to the manufactures’ instruction. However, there was no significant depletion in protein expression of SGLT2 in the siRNA group compared to the control group (Appendix Figure 1). Although we performed the siRNA experiment twice at different concentration of siRNA (5 nM and 10 nM), no significant difference was seen in protein expression of SGLT2 in the siRNA group compared to the control group (Appendix Figure 1). The reason for the unsuccessful depletion of SGLT2 protein expression remains unclear, a possible explanation is following: Although expression of mRNA for SGLT2 was decreased by siRNA treatment, expression of mRNA for SGLT2 was detected even in Hep3B cells treated with 10 nM of siRNA (Appendix Figure 2). These data suggest that a decrease in mRNA for SGLT2 was not sufficient for depletion of protein expression of SGLT2. Thus, it is difficult to examine effects of SGLT2 on metabolisms and cell proliferation by siRNA for SGLT2 in Hep3B cells at this moment. To clarify this issue, further study will be focused on an alternating approach such as short hairpin RNA, which causes more continuous suppression of target protein than siRNA [3]. This issue was described as a limitation of this study (Page 38, line 17-19).

Method for Appendix Figure 1 and 2

 Knockdown of SGLT2 in Hep3B using small interfering RNA (siRNA)

Hep3B cells were seeded into six-well plates at a density of 1×105 cells/well. The cells reached 80% confluence 48 hours after incubation and were transiently transfected with siRNA using Lipofectamine® 2000 (Invitrogen; Thermo Fisher Scientific, Inc.). The sequences for siRNA (Invitrogen; Thermo Fisher Scientific, Inc.) were as follows: Sense, 5′- CCGGAGCUGUAUUCAUCCATT-3′ and anti-sense, 5′- UGGAUGAAUACAGCUCCGGAG-3. Cell transfection was performed, according to the manufactures’ instruction. Briefly, each sequence of 10 μM　siRNA (1.5μL) and 9 μL Lipofectamine® 2000 was diluted in serum-free medium (300 μL) at room temperature for 5 min, mixed together. The mixture was subsequently administered to the Hep3B cells, then the medium was replaced with complete medium. Hep3B cells were seeded into six-well plates at a density of 5×104 cells/well and protein expression of SGLT2 was evaluated at 24, 48, 72, 96, and 120 hours after incubation.

2. Please mention if the effects are CANA specific or common features of SGLT2 inhibitors.

Answer: We appreciate for your comment. As you suggested, we examined the effect of dapagliflozin (10μM), a SGLT2 inhibiter, on cell proliferation in Hep3B. There was no significant difference in cell number between the dapagliflozin and control groups (supplementary figure 3). Moreover, Obara et al. reported that tofogliflozin, a SGLT2 inhibiter, did not suppress cell number in Huh7 and JHH cells [4]. Hung MH et al. reported that CANA specifically inhibited cell proliferation due to β-catenin-related pathway, which was not seen in the dapagliflozin and empagliflozin groups in Huh7 and Hep3B [5]. In this study, CANA-specific mechanism on inhibition of cell proliferation remains unclear; however, these previous reports along with our additional data suggest that suppression of cell proliferation in hepatoma cells may be CANA specific effect. This discussion was added in the revised manuscript (Page 37, line 16-Page38, line 2).

References

1. Ma Z, Zhu P, Shi H, Guo L, Zhang Q, Chen Y, et al. PTC-bearing mRNA elicits a genetic compensation response via Upf3a and COMPASS components. Nature. 2019;568(7751):259-63. doi: 10.1038/s41586-019-1057-y. PubMed PMID: 30944473.

2. El-Brolosy MA, Kontarakis Z, Rossi A, Kuenne C, Gunther S, Fukuda N, et al. Genetic compensation triggered by mutant mRNA degradation. Nature. 2019;568(7751):193-7. doi: 10.1038/s41586-019-1064-z. PubMed PMID: 30944477; PubMed Central PMCID: PMCPMC6707827.

3. Yang X, Wu X, Yang Y, Gu T, Hong L, Zheng E, et al. Improvement of developmental competence of cloned male pig embryos by short hairpin ribonucleic acid (shRNA) vector-based but not small interfering RNA (siRNA)-mediated RNA interference (RNAi) of Xist expression. J Reprod Dev. 2019;65(6):533-9. doi: 10.1262/jrd.2019-070. PubMed PMID: 31631092; PubMed Central PMCID: PMCPMC6923154.

4. Obara K, Shirakami Y, Maruta A, Ideta T, Miyazaki T, Kochi T, et al. Preventive effects of the sodium glucose cotransporter 2 inhibitor tofogliflozin on diethylnitrosamine-induced liver tumorigenesis in obese and diabetic mice. Oncotarget. 2017;8(35):58353-63. doi: 10.18632/oncotarget.16874. PubMed PMID: 28938561; PubMed Central PMCID: PMCPMC5601657.

5. Hung MH, Chen YL, Chen LJ, Chu PY, Hsieh FS, Tsai MH, et al. Canagliflozin inhibits growth of hepatocellular carcinoma via blocking glucose-influx-induced beta-catenin activation. Cell Death Dis. 2019;10(6):420. doi: 10.1038/s41419-019-1646-6. PubMed PMID: 31142735; PubMed Central PMCID: PMCPMC6541593.

---

## [Editor Report · Decision Letter 2]

13 Apr 2020

Effects of Canagliflozin on Growth and Metabolic Reprograming in Hepatocellular Carcinoma Cells: Multi-Omics Analysis of Metabolomics and Absolute Quantification Proteomics (iMPAQT)

PONE-D-19-33203R2

Dear Prof. Takumi Kawaguchi,

We are pleased to inform you that your manuscript has been judged scientifically suitable for publication and will be formally accepted for publication once it complies with all outstanding technical requirements.

With kind regards,

Tatsuo Kanda, M.D., Ph.D.

Academic Editor

PLOS ONE

---

## [Editor Report · Acceptance letter]

15 Apr 2020

PONE-D-19-33203R2 

Effects of Canagliflozin on Growth and Metabolic Reprograming in Hepatocellular Carcinoma Cells: Multi-Omics Analysis of Metabolomics and Absolute Quantification Proteomics (iMPAQT) 

Dear Dr. Kawaguchi:

I am pleased to inform you that your manuscript has been deemed suitable for publication in PLOS ONE. Congratulations! Your manuscript is now with our production department. 

With kind regards,

on behalf of

Dr. Tatsuo Kanda 

Academic Editor

PLOS ONE